# Multiple symptoms of total ozone recovery inside the Antarctic vortex during Austral spring

Andrea Pazmino[1], Sophie Godin-Beekmann[1], Alain Hauchecorne[1], Chantal Claud[2], Sergey Khaykin[1], Florence Goutail[1], Elian Wolfram[3,4], Jacobo Salvador[3,4,5], Eduardo Quel[3]

[1]LATMOS, UVSQ Univ. Paris Saclay, UPMC Univ. Paris 06, CNRS, Guyancourt, France
[2]LMD, CNRS, Ecole Polytechnique, Palaiseau, France
[3]CEILAP-UNIDEF (MINDEF-CONICET), UMI-IFAECI-CNRS-3351, Villa Martelli, Argentina
[4]Universidad Tecnológica Nacional, Facultad Regional Bs. As. (UTN-FRBA), Ciudad Autónoma de Bs. As., Argentina
[5]Universidad Nacional de la Patagonia Austral, Unidad Académica Río Gallegos (UNPA-UARG) and CIT-CONICET Río
Gallegos, Argentina

*Correspondence to*: Andrea Pazmino (andrea.pazmino@latmos.ipsl.fr)

**Abstract.** The long-term evolution of total ozone column inside the Antarctic polar vortex is investigated over the 1980-2017 period. Trend analyses are performed using a multilinear regression (MLR) model based on various proxies for the evaluation of ozone interannual variability (heat flux, Quasi-Biennial Oscillation, solar flux, Antarctic Oscillation and
aerosols). Annual total ozone column corresponding to the mean monthly values inside the vortex in September and during the period of maximum ozone depletion from September 15th to October 15th are used. Total ozone columns from the Multi-Sensor Reanalysis (MSR-2) dataset and from a combined record based on TOMS and OMI satellite datasets with gaps filled by MSR-2 (1993 – 1995) are considered in the study. Ozone trends are computed by a piecewise trend proxy (PWT) that includes two linear functions before and after the turnaround year in 2001 and a parabolic function to account for the
saturation of the polar ozone destruction. In order to evaluate average total ozone within the vortex, two classification methods are used, based on the potential vorticity gradient as a function of equivalent latitude. The first standard one considers this gradient at a single isentropic level (475K or 550K), while the second one uses a range of isentropic levels between 400K and 600K. The regression model includes a new proxy (GRAD) linked to the gradient of potential vorticity as a function of equivalent latitude and representing the stability of the vortex during the studied month period. The
determination coefficient ($R^2$) between observations and modeled values increases by ~0.05 when this proxy is included in the MLR model. Highest $R^2$ (0.92-0.95) and minimum residuals are obtained for the second classification method for both datasets and months periods.

Trends in September over the 2001 – 2017 period are statistically significant at 2 sigma level with values ranging between 1.84±1.03 and 2.83±1.48 DU yr$^{-1}$ depending on the methods and considered proxies. This result confirms the recent studies
of Antarctic ozone healing during that month. Trends from 2001 are 2 to 3 times smaller than before the turnaround year as expected from the response to the slowly ozone-depleting substances decrease in Polar regions.

For the first time, significant trends are found for the period of maximum ozone depletion. Estimated trends from 2001 for the 15Sept-15Oct period over 2001 – 2017 vary from 1.21±0.83 to 1.96 DU±0.99 yr$^{-1}$ and are significant at 2σ level.

MLR analysis is also applied to the Ozone Mass Deficit (OMD) metric for both periods, considering a threshold at 220 DU
and total ozone columns south of 60°S. Significant trend values are observed for all cases and periods. A decrease of OMD of 0.86±0.36 Mt yr$^{-1}$ and 0.65±0.33 Mt yr$^{-1}$ since 2001 are observed in September and 15Sept-15Oct, respectively.

Ozone recovery is also confirmed by a steady decrease of the relative area of total ozone values lower than 175 DU within the vortex in the 15Sept-15Oct period since 2010 and a delay in the occurrence of ozone levels below 125 DU since 2005.

# 1 Introduction

The evolution of total ozone content (TOC) in Antarctica during Austral spring is strongly linked to the important stratospheric ozone decline that was highlighted for the first time by Chubachi et al., 1984 and Farman et al., 1985. Nowadays the photochemical and microphysical processes leading to the massive and seasonal destruction of ozone in Polar Regions are well understood. The latest Ozone Assessment Reports (WMO, 2007, 2011, 2014) have confirmed the stabilization of ozone loss in Antarctica since 2000. The challenge now is to assess the impact of the observed reduction in the concentration of ozone depleting substances (evaluated in the polar regions to ~10 % in 2013 from the peak values in 2000, WMO, 2014) on the amplitude of the ozone destruction every year. During the last decade, several studies have been carried out to quantify a possible increase in total ozone column in the Antarctic polar vortex in spring directly linked to this decrease in the polar stratosphere. Most analyses use multi-parameter linear regression (MLR) models with different proxies to represent the interannual variability of ozone as a function of the 11 year solar cycle, the quasi-biennial oscillation (QBO), volcanic aerosols or eddy heat flux (Salby et al., 2012; Kuttipurath et al., 2013; De Laat et al., 2015). These studies generally show a significant increase of TOC since 2000 for September-November average period but they differ on the proxies used for the quantification of ozone inter-annual variability. De Laat et al. (2015) used a "big data" ensemble approach to calculate trends. Several scenarios were considered for the period over which the ozone record is calculated and for the different proxy records. They found that the significance of trends could vary from negligible to 100 % significant at 2 sigma levels depending on the scenario considered. They have also determined the optimal proxy records and ozone record scenarios to obtain the best regression. The limitation of MLR analysis is that only formal statistical error of trend is estimated and structural uncertainties linked to the single and arbitrary combination of proxies is not taken into account. De Laat et al. (2017) inferred trend values from daily Ozone Mass Deficit (OMD) computed from a multi-sensor reanalysis dataset without using any model but filtering the anomalous years with low polar stratospheric cloud (PSC) volume. The authors found positive and highly significant trend of OMD since 2000.

Solomon et al. (2016) evaluated trends in total ozone and ozone profiles records as well as healing characteristics by combining measurements (satellites and ozonesondes) and a Specified Dynamics version of Whole Atmosphere Community Climate Model (SD-WACCM). They found a significant healing in September but not in October where ozone depletion is largest during the first two weeks. They also explain the difficulty of estimating trend in October by the large variability of ozone linked to temperature variations and transport. The baroclinicity of the polar vortex in October and its displacement from the geographic pole can also contributes to the variability of the total ozone series averaged during the month of October.

The direct link observed between positive trends of total ozone within the polar vortex and the reduction of ozone-depleting substances (ODS) remains open to debate, given the natural variability of the Antarctic vortex and the possible contribution of greenhouse gases (GHGs) to the trends (Chipperfield et al., 2017).

The purpose of the present paper is to provide an update of the ozone evolution inside the Antarctic vortex during the last decades taking into account the vortex baroclinicity. The main aim is to determine the different contributions to ozone inter-annual variability and to estimate the post 2001 total ozone trend and related significance for different periods: September, which corresponds to the period of fastest development of catalytic photochemical ozone destruction and mid-September to mid-October when the maximum ozone loss is reached.

This paper is organized as follows. Ozone datasets from satellites and multi-sensor reanalysis are presented in Sect. 2 and the description of the method used for total ozone column classification inside the vortex in Sect. 3. The influence of vortex baroclinicity on total ozone column inside the vortex is assessed in Sect. 4 by using a new classification method compared to standard ones based on a single isentropic level. Ozone trends before and after a the turnaround year calculated using multi-regression model for September and mid-September to mid-October are presented and discussed in Sect. 5. Results on trends

using OMD records as a metric are also presented. The temporal evolution of the amount of very low total ozone values inside the vortex is evaluated in Sect.6. Conclusions are finally presented in Sect. 7.

## 2 Total ozone column data series

Total ozone global fields from satellite observations (TOMS, and OMI) and multi-sensor reanalysis (MSR) are used in this study to cross-check trend estimation before and after a turnaround year over the 1980-2017 period.

### 2.1 Space-borne observations

Total ozone columns data series of NASA's Total Ozone Mapping Spectrometer (TOMS) instrument onboard Nimbus-7 (N7) and Earth Probe (EP) between 1980 and 2004 are used. The instrument is a single monochromator that was designed for near-nadir measurements of the total ozone column (e.g. McPeters et al., 1998). TOMS measures the backscattering of solar radiation by the Earth's atmosphere in six 1 nm-bands of ultraviolet wavelength between 306 nm to 380 nm, more or less absorbed by ozone. Total ozone column is inferred from the ratio of two wavelengths, 317.5 nm strongly absorbed by ozone and 331.2 nm weakly absorbed (Bhartia and Wellemeyer, 2002). Level 3 gridded TV8 data of $1.0°$ (lat) $\times$ $1.25°$ (lon) of total ozone columns of TOMS were used in this work and are available from the Goddard Earth Sciences Distributed Information and Services Center (GES DISC) in simple ASCII format in the NASA anonymous ftp site (ftp://toms.gsfc.nasa.gov/pub/*satellite*/data/ozone/)

Ozone total column observations of Ozone Monitoring Instrument (OMI) onboard Aura satellite are also used to continue TOMS measurements from 2005 to 2017. The OMI instrument is a nadir viewing hyperspectral imaging in a push-broom mode. OMI measures the solar backscatter radiation in the complete spectrum of the ultraviolet/visible wavelength range (270 nm - 500 nm) with 0.5 nm spectral resolution (Levelt et al., 2006). Total ozone column used in this work was retrieved using TV8 algorithm, hereafter referred to as OMIT in order to maintain continuity with TOMS data record (McPeters et al., 2008). Level 3 daily gridded data of OMIT with better spatial resolution ($1.0°$ x $1.0°$) than TOMS is used. Data are also available on NASA anonymous ftp site.

The total ozone column data series was combined by using specific satellite data over the following periods: TOMS-N7 (1980-1992), TOMS-EP (1996-2004) and OMI (2005-2017). Note that data of 1993-1995 are sparse or missing for the September-October period. In order to complete the data series, total ozone columns of multi-sensor reanalysis 2 (see Sect. 2.2). Since TOMS and OMI UV sensors do not receive enough UV light in early September, originating from regions not illuminated by the Sun (from 77°S to 82.5°S up to mid-September), these regions were not considered to compute the total ozone mean value in MSR-2 data.

TOMS/OMI and MSR-2 data series have previously been used in different scientific studies of ozone recovery in the Southern polar region (Salby et al., 2012; Kuttipurath et al. 2013; Solomon et al., 2016). Hereafter the 1980-2017 composite satellite total ozone series will be called SAT.

### 2.2 Multi-Sensor reanalysis

Ozone Multi-Sensor Reanalysis version 2 (MSR-2) provides global assimilated ozone fields for the period 1980-2017 based on 14 satellite data sets (van der A et al., 2015). The 14 polar orbiting satellites measuring in the near-ultraviolet Huggins band were corrected to construct a merged satellite data series that are assimilated within the chemistry-transport assimilation model TM3-DAM to obtain MSR-2 data (see van der A et al., 2010 for a detailed description and van der A et al., 2015 for last improvements of the assimilation model). Corrections of offset, trends and variations of solar zenith angle and temperature in the stratosphere were computed in satellite data sets by comparisons with individual ground-based

Dobson and Brewer measurements from World Ozone and Ultraviolet Data Center (WOUDC). Those corrections are specified in van der A et al. (2015), table 2.

Daily gridded forecast ozone data of MSR-2 at 12 UTC and spatial resolution of 0.5°×0.5° were used in this work and they are available from the Tropospheric Emission Monitoring Internet Service (TEMIS) of KNMI/ESA (http://www.temis.nl/protocols/o3field/data/msr2/).

Different studies on trends in the South Hemisphere have used MSR-2 data (Kuttipurath et al., 2013, de Laat et al., 2015, 2017). Hereafter the 1980-2017 ozone series will be called MSR-2.

## 3 Data classification within the vortex

In order to consider total ozone columns only within the polar vortex, the data classification is performed by evaluating the vortex's position at different isentropic levels from May 1$^{st}$ to December 31, each year. Two classification methods are then applied in order to evaluate the impact of baroclinicity of the vortex on the averaged total ozone columns in both studied depletion periods. The first one is based on a single isentropic level, while the second one considers a range of isentropic levels.

### 3.1 Vortex position

For each day of the studied periods, the vortex position is determined by using a 2-D quasi-conservative coordinate system (equivalent latitude/potential temperature) described by McIntyre and Palmer (1984) where the pole in equivalent latitude (EL) corresponds to the position of maximum potential vorticity (PV). This conservative system is computed from PV field simulated by the Modélisation Isentrope du transport Mésoéchelle de l'Ozone Stratosphérique par Advection (MIMOSA) PV advection model (Hauchecorne et al., 2002). The model was forced by ERA-Interim (Dee et al., 2011) meteorological data (2.5°×2.5°) of European Centre for Medium-Range Weather Forecasts (ECMWF). Daily advected PV fields (1°×1°) on the 30°S–90°S latitude band at 12 UTC are used to calculate EL on the isentropic level range between 400K and 600K with a step of 25K.

Following Nash et al. (1996), PV is evaluated as a function of EL and three particular regions are identified: inside the vortex, characterized by high PV values, at the vortex edge, corresponding to high PV gradients and outside the vortex (or surf zone) with small PV values. The limit of the vortex corresponds to the EL of maximum PV gradient, weighted by the wind module. This limit is subsequently smoothed temporally with a moving average of 5 days to reduce the noise in the vortex edge data series.

### 3.2 Methodology for classification

The Nash criterion was already used in several studies to distinguish measurements (ozone profiles and total columns) inside and outside the vortex in the Southern Hemisphere (Godin et al., 2001; Bodeker et al., 2002; Pazmino et al., 2005, 2008; Kuttipurath et al., 2013, 2015). In the case of total columns, measurements were considered inside the vortex when their corresponding EL was larger than the EL of the vortex limit at a specific isentropic level (e.g. 550 K, Bodeker et al., 2002; Pazmino et al., 2005). However this "standard" method does not take into account the baroclinicity of the vortex. It can result in the classification of total ozone columns inside the vortex while partial columns below or above the selected isentropic level are outside the vortex. The total ozone column may thus not represent the ozone behaviour inside the vortex. In order to consider possible vortex baroclinicity, another approach is used, where vortex classification at different isentropic levels is considered at the same time. For this second approach, the range of selected isentropic levels is chosen in the altitude region of maximum ozone depletion: from 400 K to 600 K with a step of 25 K. The same 9 isentropic levels considered for 400 K-600 K range classification are applied each year.

In order to illustrate the impact of vortex baroclinicity on the classification of total ozone column inside the vortex, Fig. 1 shows MSR-2 total ozone fields on October 7, 2012, with the vortex position computed at different isentropic levels superimposed. The vortex position curves are represented by black to light grey colours. On this particular day, the region classified inside the vortex in the 400 K-600 K range is limited by the vortex position at 400 K (black line) towards Antarctic West coast and Palmer Peninsula and at 600 K (light grey line) towards the Antarctic East coast. The white dot marks in Fig. 1 show the limit of the region considered in this new classification. In the case of standard classification using a single level at 475 K or 550 K, the region estimated as inside the vortex consists of an area with total ozone columns larger than 400 DU. These areas are not considered in the classification using several isentropic levels between 400 K and 600 K. Regions where total ozone columns are lower than 220 DU are taken into account by the classification at all the isentropic levels. A daily mean total ozone column of 213.4 DU was computed inside the vortex using this new classification method. The standard classification estimates a 40 DU and 20 DU larger ozone average values at 475 K and 550 K respectively on that day.

## 4 Vortex baroclinicity

Both methods of classification described in the previous section were applied to satellite composite total ozone data series SAT and MSR-2 reanalysis at each grid point. For each year, daily mean total ozone amount inside the vortex was averaged over two periods: the whole month of September, and the period of maximum ozone depletion between September 15[th] and October 15[th] (15Sept-15Oct). Figure 2 shows the evolution of total ozone average inside the vortex for the 15Sept-15Oct period between 1980 and 2017 for the MSR-2 data series computed with the standard classification method based on the single isentropic level (475 K and 550 K) and with the second method using the 400 K – 600 K range of isentropic levels. Error bars represent the two sigma standard error ($2\sigma$). Similar interannual total ozone variability is observed for the time series obtained by the different methods. The correlation coefficients between the range method and the standard one at 475 K and 550 K are 0.98 and 0.99 respectively. Despite these good correlations, the data series are significantly different at the $2\sigma$ level. Larger ozone values are found with the standard method, especially for the 475 K level, which shows a mean difference with the TOC ozone time series based on the range method of ~15 % over the whole analysis period. Three years stand out in the comparison: 1995, 1999 and 2011, during which the inside vortex region was systematically larger at 475 K compared to higher isentropic levels during the period. Similar results are observed for September (not shown). In this work, the second method is preferred since it takes into account the ozone loss at different isentropic levels, which strongly impacts the total column.

MSR-2 total ozone time series obtained in September and 15Sep-15Oct with the range classification method are displayed in Figure 3. September presents ~8% larger ozone mean values than the 15Sept-15Oct period. Similar interannual variability is observed between the two periods as shown by the correlation coefficient of 0.98. The last four years present very similar ozone values around 205 DU in September while in 15Sept-15Oct period they show larger variability.

MSR-2 total ozone data series inside the vortex are compared to SAT series as shown in Fig. 4, which displays the relative difference between MSR-2 and SAT for the 400 K-600 K range classification. Differences of about ±0.5 % are observed in the 1980s. Small differences are expected during this period since only TOMS data are used in both data sets until 1993. In the 1993-1995 period discrepancies between both curves are only due to the differences in the selection of MSR-2 data for the SAT record in order to have similar spatial coverage as the data from the other instruments incorporated in the SAT time series. These differences varying between -1 and 0.5 % represent an estimation of the impact of reduced spatial coverage in SAT dataset on the averaged total ozone level in September. The 15Sept-15Oct period presents negligible differences. The addition of GOME (1996-2005) in MSR-2 assimilation could explain the discrepancies with the SAT dataset that considers only TOMS-EP. From 2001, differences are larger and generally positive, reaching ~5% in September and ~3% in 15Sept-15Oct. period. These increased differences are especially visible during the period where data from instruments on board the

ENVISAT platform (e.g. SCIAMACHY) are assimilated in the MSR-2 record. Overall, values in September present a mean bias of 1.3 % (dash blue line in Fig. 4), and in 15Sept-15Oct a smaller bias value of 0.5 % (dash red line in Fig. 4). Temporal evolution of the differences, e.g. negative trend in the 1980s and positive trend in the 2000s, can have an impact on the long-term ozone trends retrieved from both records. In general, differences between SAT and MSR-2 records are caused by MSR-2 starting to use multiple satellite total ozone columns records after 1996, the procedures in MSR-2 to account for inter-instrument differences, and the data assimilation methodology that allows for filling gaps (van der A et al., 2015).

Despite the differences between those datasets, a purpose of this work was to analyse in the same way satellite data such as those included in the SAT record without any correction or adjustment and the MSR-2 record, which accounts for inter-instrument differences using ground-based total column data. Due to the larger differences observed between both data sets in September especially in the 1995 – 2010 period, which may have an impact on trend analysis, it was decided to retrieve trends from the SAT dataset in the 15Sept-15Oct only.

In the next section, ozone data series based on the different classification methods are used to evaluate the impact of vortex baroclinicity on ozone trends inside the vortex for both studied periods.

## 5 Trend analysis

### 5.1 Method

In order to evaluate ozone recovery in Antarctica, estimation of trends before and after 2001 were calculated using a multi-regression model (Nair et al., 2013) updated from the AMOUNTS (Adaptative Model for Unambiguous Trend Survey) model (Hauchecorne et al., 1991; Kerzenmacher et al., 2006). Different common explanatory variables such as eddy heat flux (HF), solar flux (SF), Quasi-Biennial Oscillation (QBO), Aerosols (Aer), Antarctic Oscillation (AAO) are used to explain total ozone variability over the 1980-2017 period. These proxies were widely applied in different trend studies (e.g. de Laat et al., 2015 and references herein). The ODSs contribution to long-term trend in ozone is represented by piece wise trend functions (PWT). The total ozone variability ($Y$) can be expressed following Eq. (1):

$$Y(t) =$$
$$K + C_{HF}\text{HF}(t) + C_{SF}\text{SF}(t) + C_{QBO30}\text{QBO30}(t) + C_{QBO10}\text{QBO10}(t) + C_{Aer}\text{Aer}(t) + C_{AAO}\text{AAO}(t) + C_{GRAD}\text{GRAD}(t) +$$
$$\text{PWT}(t) + \in (t), \tag{1}$$

where t is year from 1980 to 2017, K is a constant, $C_{proxy}$ are the regression coefficients of the respective proxies mentioned above and $\in (t)$ is the total ozone residuals. Table 1 shows the respective information for each proxy: source, specific characteristics and time window where proxy values are averaged to represent the respective year value. QBO effect on ozone variability is estimated using two proxies at 30hPa (QBO30) and 10hPa (QBO10), which are out of phase by $\sim \frac{\pi}{2}$ (Steinbrecht et al., 2003). The HF proxy corresponds to the average over the August-September period of the 45-day mean Heat Flux in the 45°S-75°S latitude range at 70hPa from MERRA-2 analyses. The time window of August-September is selected for computing the mean HF, following de Laat et al. (2015) recommendation to obtain the best regression results. For the Aer term, a merged proxy of monthly aerosol optical depth (AOD) is computed from updated Sato et al., (1993) dataset for the 1980-1990 period and from four satellite data series (SAGE II, OSIRIS, CALIOP and OMPS) for the 1991-2017 period. AOD datasets are averaged over the 40°S-65°S zonal region in the 15-30 km altitude range. Updated Sato et al. data are obtained from NASA monthly AOD at 550nm. The satellite AOD data over 1991-2017 period were computed at 532 nm. The Sato et al. data set was converted to 532 nm according to Khaykin et al. (2017). The merged AOD proxy was obtained by normalizing the Sato et al. time series to the SAGE II data in December 1991. The regression code uses the AOD values in April before the complete formation of the vortex in order to avoid possible contamination of aerosols

satellite data by Polar Stratospheric Clouds. The April AOD proxy is represented by a bold black line in Fig. 5 together with Sato et al. (1993) and satellites datasets for the 1991-2017 period.

A new GRAD(t) proxy was developed in order to take into account the stability of the vortex during the studied period. This proxy corresponds to the maximum gradient of PV as a function of EL at 550 K during both studied periods (e.g. September and 15Sept-15Oct). It is calculated from ERA-Interim data. GRAD and HF proxies are detrended by removing a $3^{rd}$ order polynomial fit to minimize correlation with PWT proxies. Figure 6 displays GRAD and HF proxies before and after removing trends. An anti-correlation of ~0.55 between these two proxies is observed with a p-value <0.01, but the addition of GRAD proxy provides a much better agreement between measurements and model, especially during the last decade. The contribution of the GRAD(t) proxy to the improvement of the MLR results is discussed in Sect. 5.2.3.

For the long-term trends, two piecewise linear trend ($PWLT(t) = C_{t1}t1(t) + C_{t2}t2(t)$) functions calculated before and after the turnaround year are usually used to estimate the change of slope in the long-term evolution of ozone due to ODS (e.g. Reinsel et al., 2002; Kuttipurath et al., 2013, de Laat et al., 2015). In this work our Modified PWLT model (PWT) uses an additional function in order to take into account the slower growth of ODS near the turnaround year and the ozone loss saturation effect within the Antarctic polar vortex in October (Yang et al., 2008). The PWT model is represented by Eq. (2):

$$PWT(t) = C_{t11}t11(t) + C_{t12}t12(t) + C_{t2}t2(t) \tag{2}$$

where $C_{t11}$ and $C_{t2}$ are the coefficients of the linear functions and $C_{t12}$ of the parabolic function. The first period is represented by a linear time proxy t11 and a parabolic time proxy t12. The second period is expressed only by a linear time proxy t2. The proxies are computed as follow:

$$t11 = \begin{cases} t & 0 < t \le T_0 \\ T_0 & T_0 < t \le T_{end} \end{cases} \tag{3}$$

$$t12 = \begin{cases} \left(t - \left(\frac{T_0+1}{2}\right)\right)^2 & 0 < t \le T_0 \\ \left(\frac{T_0-1}{2}\right)^2 & T_0 < t \le T_{end} \end{cases} \tag{4}$$

$$t2 = \begin{cases} 0 & 0 < t \le T_0 \\ t - T_0 & T_0 < t \le T_{end} \end{cases} \tag{5}$$

$T_0$ corresponds to the turnaround year in the considered period. In this work, 2001 was selected as the turnaround year when equivalent effective stratospheric chlorine (EESC) maximizes for a mean age-of-air of 5.5 yr (Newman et al., 2007). The corresponding value for $T_0$ is 22. $T_{end}$ corresponds to the number of years considered in the study (38 for 1980-2017). The minimum of the parabolic time proxy t12 is set to the middle of the period before the turnaround year so that the slope of the proxy is zero on that year. In this case the coefficient of t11 ($C_{t11}$) can be considered as the linear trend before 2001. After 2001, t11 and t12 are constant and then the linear trend is given by the $C_{t2}$ coefficient. Figure 7 represents the evolution of the three piece-wise proxy anomalies normalised by the corresponding standard deviation. The improvement using PWT instead of PWLT is discussed in Sect. 5.2.4.

## 5.2 Trend results for the averaged total ozone column records

The multi-regression model described in previous section was applied to MSR-2 total ozone anomalies time series computed as monthly total ozone – mean total ozone for the September and 15Sept-15Oct periods and to SAT for the 15Sept-15Oct period only. Times series of total ozone data corresponding to the different classification methods described in Sect. 4 were also used to evaluate the impact of vortex baroclinicity on total ozone trends.

### 5.2.1 September

A rapid decrease of ozone levels occurs within the polar vortex in Antarctica from the last two weeks of August to the end of September when the necessary sunlight to start the ozone catalytic destruction cycles is present again above austral polar regions. Important differences in total ozone levels are found inside the vortex between the first and second half of September, with very low values observed mostly during the last week. Although pronounced decrease in total ozone is observed in September, recent publications have used ozone records obtained during this month to detect the ozone recovery (Solomon et al., 2016; Chipperfield et al., 2017; Weber et al., 2017). Those studies use data and/or simulations poleward of ~60°S and identify first signs of Antarctic ozone recovery for September but not yet for October due to the larger dynamical variability during that month. In this paper, results from our multi-regression model are evaluated and compared to those previous publications for the September period. Figure 8 illustrates the results of the regressions model described in section 5.1 for the MSR-2 total ozone data series inside the vortex using the 400 K-600 K range classification. The top panel represents the deseasonalized total ozone observations as well as the regressed ozone values. The model results reproduce quite well the interannual variability of measurements except in 2002 when the vortex split in two parts in late-September due to a major sudden stratospheric warming (e.g., Allen et al., 2003). Likewise, the year 2000 was characterized by a large ozone hole area in September, and yields a relatively high value of residual of ~30 DU on that year. Contributions of the different proxies are shown in the second to fourth panels of Fig. 7. Fitted HF and GRAD were added (black line in second panel of Fig. 8) due to the correlation between both proxies. The model term linked to the HF+GRAD fitted proxy represents the second largest contribution to total ozone interannual variability (~10 % of the total variance) after the PWT proxy which contributes to about 80 % of the total variability. Other proxies (third panel of Fig. 8) represent only 1 % of total ozone variability. Aerosol proxy contributes by respectively ~6 DU in 1992 and ~3 DU in 1983 due to Pinatubo and El Chichon eruptions. Negligible impact is seen in other years. Fitted QBO (QBO30hPa + QBO10hPa) explains ±5 DU ozone variability. The contributions of SF and AAO proxies are negligible.

The model explains 92 % of the ozone variability as deduced from the determination coefficient $R^2$. The estimated total ozone trends before and after 2001 are -5.31±0.67 DU yr$^{-1}$ (-25.2±3.2 % decade$^{-1}$) and 1.84±1.03 DU yr$^{-1}$ (8.8±4.9 % decade$^{-1}$), respectively. Both trends are significant (i.e. statistically different from zero) at 2σ. The 1980-2000 period presents larger depletion rate compared to Weber et al., 2017 (from -12 to -19% per decade depending on dataset) and comparable rate for the period of recovery (8-10 % decade$^{-1}$). Comparable values of trends are found when the 475 K classification level is used (-21±3.2 % decade$^{-1}$ and 10.1±5 % decade$^{-1}$). The 400 K-600 K classification allows us to obtain the best agreement between observations and regressed values (larger $R^2$) and lower $\chi$ ($\sqrt{\sum_i(obs_i - mod_i)^2/(n-m)}$)) of residuals. Those results are represented in Table 2 for MSR-2 total ozone datasets inside the vortex and for the three classifications analysed in this study. Despite trend values after 2001 for the 475 K classification are larger by about 28 % than for the 400 K-600 K classification range, trend results between both classifications are not significantly different at 2σ level, suggesting a limited effect of vortex baroclinicity on trend estimation using MLR analysis. The different results in Table 2 generally present a ratio between trends before and after 2001 close to 3, similar to that of ODS trends before and after the peak (Chipperfield et al., 2017). This indicates that the ozone recovery trend could be due to ODS decrease. Nonetheless this trend cannot be reliably associated to chemical processes alone and other processes could also play a role.

Computed trends over the 2001-2017 September period obtained with our model range from 1.84 to 2.36 DU yr$^{-1}$ for all cases studied. They are all significant at 2σ level. Solomon et al. (2016) found significant total ozone trend of 2.5±1.7 DU yr$^{-1}$ in September from SBUV and ozonesonde observations and similar results from the chemistry+dynamics+volcanoes (Chem-Dyn-Vol) simulation (2.8±1.6 DU yr$^{-1}$) using the Whole-Atmosphere Community Climate Model (WACCM). Estimated total ozone trend when only chemistry is considered in the model (Chem-Only) correspond to only half of the final trend (1.3±0.1 DU yr$^{-1}$).

A simulation test was done to evaluate the pertinence of using other proxies than PWT, HF and GRAD since only these fitted proxies present significant regression coefficient values at 95 % confidence interval. Results are represented in Table 2. Slightly lower determination factor $R^2$ is computed if only PWT, HF and GRAD are considered for September and comparable residual and trends. This results suggest that the others proxies provide marginal improvement to the MLR

analysis.

### 5.2.2 September 15 to October 15

In order to confirm healing of the Antarctic ozone hole, it is important to evaluate trends for the period where lowest total ozone values are observed inside the vortex e.g. between September 15[th] and October 15[th]. The same analysis as for September is thus performed. Figure 9 illustrates the results of regressions model for total ozone of MSR-2 data series inside

the vortex using the 400 K-600 K classification. It shows that the interannual variability of measurements is better represented by the model than in September. For that period, the determination coefficient $R^2$ is 0.95 (see also Table 3). As for the September regression, the sum of fitted HF and GRAD proxies (black line in second panel) represents the second largest contribution to total ozone interannual variability (~13 % of the total variance) after the PWT proxy (~80 %) and the last decade of measurement is correctly reproduced by the model. Significant trends of –5.81±0.6 DU yr$^{-1}$ (-29.8±3 %

decade$^{-1}$) and 1.42±0.92 DU yr$^{-1}$ (7.3±4.7 % decade$^{-1}$) are estimated before and after 2001. Similar results are observed if a single level classification is used with larger trend values after 2001 for 475 K. All trend results are comparable within ±2σ. Results based on the SAT record are similar with slightly larger trend values after 2001. Note that the addition in the MLR analysis of the 2 most recent years (2016-2017), which were characterized by weak ozone holes, changed the significance of the 2001-2017 trend from hardly significant to significant better than 2σ. Results obtained in the 1980-2017 period by the

MLR analysis show thus for the first time a significant recovery in the 15Sept-15Oct period. Solar Flux, QBO, Antarctic Oscillation and Aerosol (third panel of Fig. 9) explain ~1 % of the total variance. QBO explains ±3 DU interannual variability and Aerosol signal amount to ~6 DU and ~3 DU linked to Pinatubo in 1992 and El Chichon in 1983. SF contribution varies from 4.5 DU during the maximum (except for the last solar cycle, ~1 DU) to -2.2 DU during the minimum. AAO represents negligible contribution. Same test as for September was performed where proxies of SF, QBO,

AAO and Aerosol were removed from the linear regression. Results are presented in Table 3. Negligible difference in trends, $R^2$ and residuals are observed if those proxies are considered or not in the MLR analysis. In addition, lower chi-values are found for smaller number of fitted parameters, which is the case for the regression using PWT, HF and GRAD only.

The different cases shown in Table 3 present significant trends at 2σ over the 1980-2000 and the 2001-2017 periods. Computed trend with 400 K-600 K range classification is comparable to the Chem-Only trend calculated by WACCM in

Solomon et al. (2016). Despite the good agreement between regressed values and measurements especially for the period 15Sept-15Oct and for the range classification method (400 K-600 K), it is not possible to attribute ozone significant increase to ODS decrease. In addition, the ratio between trends before and after 2001 is larger than 3-which could be due to the effect of desaturation of the ozone loss.

### 5.2.3 Impact of GRAD proxy on trend estimation

The HF proxy represents the cumulative effect of wave activity on vortex stability (e.g. a high HF corresponds to a warmer vortex) that seems insufficient to represent total ozone variability over the last decade, especially in 2010 and 2012. The GRAD proxy was developed in order to consider also the vortex stability during both studied periods. Since Aerosols, QBO, SF and AAO represent lower contribution to ozone variability, trend analyses using HF, PWT proxies only and including or not the GRAD proxy are performed in order to highlight the impact of this parameter. Figure 10 shows residuals of MLR

analysis with and without GRAD on MSR-2 data inside the vortex for the 400 K-600 K classification range for September and 15Sept-15Oct periods. The residual anomalies are significantly reduced after 2002 when GRAD is used, especially in

the 15Sept-15Oct period. The second panels of Figures 8 and 9, show that in some years HF and GRAD proxies are in phase as during 2009-2014 when GRAD intensifies the HF contribution to ozone variability. This improvement is especially visible for the years 2010 and 2012. When both proxies are anticorrelated, as in 2005-2008, the improvement linked to the GRAD proxy is also observed. Table 2 and 3 show the results of the regressions excluding GRAD proxy for September and

15Sept-15Oct respectively. The determination coefficient is generally reduced by ~0.07 and the $\chi$ values are 25 % to 50 % larger. Trend values are mostly similar but the error bars are reduced when GRAD is used as explanatory variable, especially after 2001. Trends over the 2001-2017 period estimated without the GRAD proxy are still significant at $2\sigma$ in September and 15Sept-15Oct for both datasets.

### 5.2.4 PWT vs PWLT

In order to evaluate the improvement of an additional parabolic function to the linear functions of the piece-wise trend proxy, the classical piece-wise linear trends (PWLT) is applied in the MLR analysis of MSR-2 datasets. Figure S1 shows average total ozone anomalies of MSR-2 inside the vortex (400 K-600 K range classification method) in September and 15Sept-15Oct and the retrieved trends using both the PWLT and PWT methods. In the case of the 15Sept-15Oct period, the PWT model provide a better representation of long-term ozone evolution compared to PWLT, as it better captures ozone loss

saturation during the 1990s. The trends error bars are also smaller using PWT before and after 2001. In addition, a better agreement between measurements and model values is observed with a larger $R^2$ and lower residuals. The 2001-2017 trend error bars are ~60 % larger if PWLT is used and the trend value itself is nearly double. In the case of September, a slight improvement in $R^2$, residuals and error bars is obtained with PWT. The 2001-2017 trend value with PWLT is 40 % larger.

### 5.3 Results using OMD metric

OMD has been used in previous studies to evaluate ozone loss and ozone recovery (e.g. de Laat et al., 2017). This metric has the advantage to be independent of the vortex position. Total ozone MSR-2 data were used to compute the average daily OMD on September and 15Sept-15Oct periods. The total ozone columns are referenced to the 220 DU threshold value and the corresponding mass deficit of the partial column (220 DU – total ozone column) is computed at each grid point (e. g., Bodeker and Scourfield, 1995). Only total ozone columns south of 60°S and lower than 220 DU are considered and the daily

OMD correspond to the sum of OMD at each pixel multiplied by the cosine of the latitude and the square of the Earth's radius. Table 4 shows the MLR analysis of OMD using different sets of proxies as for ozone average in Table 2 and 3. The contributions of Aerosols, AAO, QBO and SF do not shown an impact on MLR analysis where similar $R^2$, $\chi$, trend and error bars values are obtained. On the other hand, the inclusion of GRAD results in larger $R^2$ and lower residuals in both periods. For the different cases and periods shown in Table 4, the OMD trend values are significant at $2\sigma$. The MLR analysis using

GRAD, HF and PWT proxies provides trends of -1.29±0.24 Mt yr$^{-1}$ and 0.86±0.36 Mt yr$^{-1}$ in September and -1.61±0.22 Mt yr$^{-1}$ and 0.65±0.33 Mt yr$^{-1}$ in 15Sept-15Oct. De Laat el al. (2017) found a similar trend for the recovery period of 0.77 Mt yr$^{-1}$ for the averaged OMD between the 220 and 280 day of year. Figure 11 displays the comparison between the OMD records and results of MLR analysis for the September and 15Sept-15Oct period, together with the trend components of the model. The effect of ozone loss saturation is particularly visible in the 15Sept-15Oct period. There are some years that are

not totally explained by the model, e.g. 2002 and 2004 for both periods and 2000 for September. The contributions of GRAD, HF and GRAD+HF are shown in upper panels of Fig. S2 where GRAD intensifies HF contribution in 2010 and 2012, while both proxies are anti-correlated in 2005-2008 as observed for the total ozone analysis. The residuals with and without GRAD are shown in the bottom panels of Fig. S2. The improvement linked to the use of GRAD proxy is particularly visible in the last decade.

As for total ozone, MLR analysis using PWLT was performed for comparison with the PWT model. Figure S3 shows the OMD records together with PWLT and PWT components of the regression model for the both periods. Similar agreement is

obtained for September but the regression results in larger residuals for 15Sept-15Oct using PWLT (not shown). Major difference is observed in the period 2001 - 2017 with a large trend value of -0.91±0.41 Mt yr$^{-1}$, corresponding to an increase of 40 % in absolute value.

## 6. Temporal evolution of low total ozone values inside the vortex

The ozone hole is generally defined as the region with total ozone columns lower than 220 DU. This standard value was used in different studies to evaluate the ozone depletion from the Ozone Hole Area (OHA) (e.g. Newman et al., 2006; Solomon et al., 2016) or the Ozone Mass Deficit (OMD) (e.g. de Laat and van Weele, 2011, de Laat et al., 2017) metrics. In order to evaluate how the ozone hole is influenced by very low ozone values, the surface relative to the vortex area occupied by ozone values lower than different threshold levels is computed for each day and averaged over different periods (September, 15Sept-15Oct and October). The top panel of Fig. 12 shows the evolution of these average relative areas for five different thresholds: 220 DU, 200 DU, 175 DU, 150 DU and 125 DU, for the 15Sept-15Oct period. MSR-2 datasets is used for this analysis and vortex areas are estimated using the 400 K-600 K range classification, the results of previous sections having shown that the range classification better constrains the ozone hole area. Results show increasing areas during the 1980s, a stabilisation in the 1990s and a larger inter-annual variability since 2001. In contrast to the 220 DU threshold case, the evolution of relative areas corresponding to lower thresholds shows a delayed increase from the beginning of the1980s to the early 1990s, reaching a maximum in all cases in 2000. After 2000, a larger interannual variability is generally observed and from 2006 a steady decrease is seen for thresholds lower than 200 DU. In all cases, several anomalous years are observed with important reduction of ozone depletion: 1988, 1991, 2002, 2004, 2010 and 2012. Note that these years correspond to a high contribution of HF+GRAD proxies to the regressed ozone values (Fig. 10, second panel). If we exclude these anomalous years, the 220 DU relative area remains fairly stable at about 90% of the total vortex in average since 1990. In the most recent years, relative areas for 125 DU and 150 DU thresholds decrease to less than 10 % and 30 % respectively from their peak value of 21 % and 57 % reached in 2000. If such trend persists, the frequency of very low ozone values (e.g. below 125 DU) is expected to become negligible in the coming decade.

In addition, OMD were computed for the same thresholds (bottom panel of Fig; 10). The evolutions of OMD present similar behaviour as the relative area but in this case, OMD at 220 DU threshold shows a visible decrease since 2000. Nowadays OMD of threshold lower than 150 DU presents very small values lower than 0.2 Mt.

Solomon et al. (2016) have highlighted for the first time a delay in the formation of the ozone hole after 2000. This shift can be explained by the slower ozone loss rates after sun appearance over the Pole, due to ODS decrease in the polar stratosphere. In this work, such possible time shift was investigated by computing the first day when ozone levels below certain thresholds occur inside the vortex (using the 400 K-600 K classification range), from September 1$^{st}$ to October 15$^{th}$ (Fig. 13). The same thresholds values as for Fig. 12 were used. In order to avoid influence of spurious values, the number of $1° \times 1°$ grid cells with total ozone columns below the various thresholds in the first day (or start day) has to be larger than 10. For each curve, day values equal to 244/245 correspond to years when ozone levels below the corresponding threshold have appeared at or before the beginning of September. For the 220 DU threshold, the dark blue curve shows that this is the case since 1983. For the 200 DU threshold, lower ozone values appear before the beginning of September after the mid-1980s. For the other thresholds, we observe a decrease, with some variability, of the start day during the 1980s, and the 1990s for the two lowest thresholds and an increase after 2000 - 2005. This increase is most visible on the 125 DU threshold curve and to some extent also on the 150 DU threshold curve. In 2016, ozone levels below 150 DU have appeared in the beginning of September, typically as for ozone holes at the end of the 1990s but levels below 125 DU still appear later. No values for a particular year in the threshold curves indicate that total ozone levels were above that threshold during the whole

period considered. This is the case for the two lower thresholds before 1985 and for the 125 DU threshold in 2002, 2004 and 2017.

**7 Conclusions**

MSR-2 and SAT (TOMS/OMI with gaps in 1993-1995 filled by MSR-2) datasets have been used to evaluate total ozone
trends within the Southern polar vortex over the 1980-2017 period. A multi-regression model is applied to ozone values averaged over the September month and the 15 September to 15 October period in order to compute long-term trends before and after the ODS peak in the polar stratosphere that occurred around 2001 (Newman et al., 2007). The 15 Sept. – 15 Oct. time range corresponds to the period of maximum ozone depletion. It is not commonly used in previous works. Proxies and time windows for averaging them are selected following de Laat et al. (2015) work.
For the classification of total ozone measurements inside the vortex , the classical Nash et al. (1996) method is used. In order to evaluate the impact of vortex baroclinicity on trend analysis, classifications using a single isentropic levels (475 K, 550 K) and a range of levels (400 K – 600 K) are tested. Systematic differences are found between the various. However the inter-annual variability is similar with correlation coefficients ranging from 0.98 to 0.99 in both studied periods. While larger trend values are generally found with the 475 K classification, the differences with trends related to the 400 K – 600 K range
classification are not significant at 2σ level.

The use of combined piece-wise linear and parabolic functions for the trend proxies (PWT) in the 1980 – 2000 and 2001 – 2017 periods provides a good representation of the total ozone long term behaviour inside the vortex (after removal of interannual variability), especially for the 15Sept-15Oct period, probably in relation with the effect of ozone loss saturation. The classical PWLT used in previous studies seems to overestimate the trends during the recovery period.
A new proxy (GRAD) representing the vortex stability over both studied periods is included in the multilinear regression. This proxy improves the representation of total ozone inter-annual variability by the regressed values especially over the last decade. It results in ~0.05 larger value for the $R^2$ determination coefficient, lower fitted residuals and smaller trend uncertainties for the different classification methods and datasets. In general, the best agreement between observations and regressed values is found for the 15Sept-15Oct period. While the HF combined with GRAD proxies reproduce quite well the
interannual variability of ozone, other proxies such as Aerosols, QBO, SF and AAO present smaller explanatory power and contribute less to reduce trend uncertainties.

In the period of increasing ODS (1980-2000), the MLR analysis shows negative and significant trends for both studied periods, similar to values found in previous studies (e.g. Kuttippurath et al., 2013 and de Laat et al., 2015). The 15Sept-15Oct period presents slightly larger negative trends in absolute value than the month of September.
In the 2001-2017 period, positive trends are obtained for all scenarios. The largest trends and highest significance are found for the September period, with a trend value of 1.84 ± 1 DU for the MSR-2 total ozone record using the 400 K-600 K range classification method. For the 15Sept-15Oct period, a lower trend of 1.42 ± 0.92 DU is obtained using the same record. Better fit and smaller residuals are obtained for that period. Differences with trend results from the other SAT data set evaluated in the study are not statistically significant.
The ratio between trends before and after 2001 varies according to the studied period. Only September trends present a ratio of ~3 as expected for an ozone response to ODS evolution. However, as for other trend studies based on MLR fit to observations, it is not possible from this analysis alone to fully attribute the retrieved trends to ODS evolution. For such a study, a combination of model and observations is needed. Potential feedbacks between chemistry, radiation and dynamics will play a role in ozone recovery. A recent study indicates an increase in temperatures within the vortex core from MERRA
reanalyses during the period 2000 – 2014 in austral spring and summer (Solomon et al., 2017). Such a temperature increase

that could be linked to ozone increase could play a role in the decrease of occurrence of low ozone values within the vortex and subsequent ozone increase.

The evolution of ozone mass deficit was also analysed using MSR-2 data. MLR analysis on this metric confirms the findings obtained for total ozone columns, e.g. a general improvement of the fits with the GRAD proxy and the main explanatory

power provided by the GRAD, HF and PWT proxies. The 2001-2017 OMD trends are larger in absolute value for September ($-0.86\pm0.36$ Mt.yr$^{-1}$) than for 15Sept-15Oct ($-0.65\pm0.33$ Mt.yr$^{-1}$). They are significant at $2\sigma$ level in both cases. These results are in general good agreement with those obtained in de Laat et al. (2017). Similar reductions of 53 % and 35 % of OMD are computed for September and 15Sept-15Oct respectively. This is consistent with the 30 % - 40 % change in ODS relative to their level in 1980 when total ozone values below the 220 DU threshold started to appear systematically (WMO, 2011).

The structural uncertainties of the MLR analysis linked to the selection of proxies were not fully analysed in this work, as in De Laat et al. (2015). The main sensitivity tests concerned the baroclinicity of the vortex and the impact of its stability during the studied periods. Trend differences in the various scenarios analysed provide some quantification of related uncertainties and are lower than the statistical trend uncertainties. Further, the large determination coefficients obtained for both periods analysed give confidence in the retrieved trends. The Heat Flux proxy that provides the largest explanatory

power in the various fits is a well-known driver of vortex temperature conditions that are the primary causes of polar ozone depletion in periods of high ODS levels. The influence of the GRAD proxy in recent years highlights the importance of the vortex stability for the containment of the ozone hole during the period of maximum depletion.

Polar ozone recovery was also evaluated by examining the temporal evolution of relative areas occupied by ozone levels below various thresholds within the vortex. Very small total ozone columns (<150 DU) did not occur inside the vortex

before the late 1980s and early 1990s. For the 125 DU, 150 DU and 175 DU thresholds, relative areas display a steady decrease since the beginning of the 21$^{st}$ century, while for the 200 DU and 220 DU thresholds, the relative area's evolution is quite stable. All relative area curves are marked by increased variability since 2000. Relative areas related to the lowest thresholds show a more rapid decrease, which further points towards polar ozone recovery. OMD records based on the same thresholds show a similar behaviour.

In summary, this work present clear symptoms of polar ozone recovery. Recovery is found for the month of September and for the first time for the period of maximum ozone depletion, e.g. from September 15 to October 15. For both studied periods, recovery is deduced from the significant positive trends in total ozone, significant negative trends of ozone mass deficit and from the steady decrease of the occurrence of low ozone values within the polar vortex. As ODS continue to decrease in the next years, it is likely that ozone recovery in the Polar vortex in spring will become more evident.

*Data availability*. The source of the different total ozone column datasets and classical proxy time series utilised in this work are publically available from the websites given in the text and in Table 1. The satellite data used to build the Aerosol proxy are available at https://eosweb.larc.nasa.gov/project/calipso/calipso_table for CALIPSO; https://eosweb.larc.nasa.gov/project/sage2/sage2_table for SAGE II, http://odin-osiris.usask.ca/ for OSIRIS; http://mls.jpl.nasa.gov/products/h2o_product.php for MLS; and

35 https://ozoneaq.gsfc.nasa.gov/data/omps/ for OMPS V1 LP. Other data as equivalent latitude and GRAD proxy are available upon request.

*Competing interests*. The authors declare that they have no conflict of interest.

*Acknowledgements*. The authors thank NASA/GSFC and TEMIS for total ozone column data of TOMS/SBUV/OMI-TOMS and MSR-2,
respectively. They are grateful to Cathy Boone of ESPRI data centre of Institut Pierre Simone Laplace (IPSL) to provide ERA interim data. This work was supported by the Dynozpol/LEFE project funded by the French Institut National des Sciences de l'Univers (INSU) of the Centre National de la Recherche Scientifique (CNRS). The authors thank Susan Solomon for fruitful interactions and on this article two anonymous referees for their constructive reviews.

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

**Table 1: Information of proxies (source, characteristics and time window for the mean yearly value).**

| Proxy | Source | Characteristics | Time window |
|---|---|---|---|
| HF | NASA/Goddard Space Flight Center https://acd-ext.gsfc.nasa.gov/Data_services/met/ann_data.html | 45-Day Mean Heat Flux between 45°S-75°S at 70hPa from MERRA 2 | Aug.-Sept. |
| SF | Dominion Radio Astrophysical Observatory (National Research Council Canada) ftp://ftp.geolab.nrcan.gc.ca/data/solar_flux/monthly_averages/solflux_monthly_average.txt | Monthly averages of Solar Flux at 10.7cm wavelength | Sept. |
| QBO | Institute of Meteorology (Freie Universität Berlin) http://www.geo.fu-berlin.de/en/met/ag/strat/produkte/qbo | Monthly mean Quasi-Biennial Oscillation at 30 and 10hPa | Sept. |
| Aer | 1980-1990: NASA/Goddard Space Flight Center https://data.giss.nasa.gov/modelforce/strataer/ | AOD@550nm, 15-30km, 40°S-65°S zonal mean. | April |
| | Jan. 1991 –April 2017 composite data series | AOD@532nm merged satellite time series of SAGE II, OSIRIS, CALIOP and OMPS following method described in Khaykin et al. (2017) 15-30km, 40°S-65°S zonal mean | |
| AAO | NOAA/National Weather Service ftp://ftp.cpc.ncep.noaa.gov/cwlinks/ | Daily AAO index | Same as O3 |
| GRAD | | Daily maximum of PV slope at 550K computed from ERA-Interim data | Same as O3 |

**Table 2: Coefficient of determination R$^2$, trends $\pm$ 2$\sigma$ in DU yr$^{-1}$ before and after the turnaround year 2001 derived from multi-regression model using as input MSR-2 (1980-2017) total ozone anomalies inside the vortex for September using three classification methods described in Sect. 3.2. The residual is represented in DU by $\chi = \sqrt{\sum_i (obs_i - mod_i)^2 / (n - m)}$ where $obs_i$ and $mod_i$ correspond to observations and model monthly mean, n the number of years and m the number of parameters fitted as in Weber et al. (2017)**

| | Multi-Sensor Reanalysis (MSR-2) | | |
|---|---|---|---|
| | 400 K-600 K | 475 K | 550 K |
| R$^2$ | 0.92 | 0.90 | 0.92 |
| Trend before 2001 | -5.31±0.67 | -4.90±0.74 | -5.23±0.68 |
| Trend after 2001 | 1.84±1.03 | 2.36±1.16 | 1.92±1.07 |
| $\chi$ | 10.74 | 12.02 | 11.12 |
| | Only with GRAD, HF and PWT | | |
| R$^2$ | 0.91 | 0.89 | 0.89 |
| Trend before 2001 | -5.32±0.64 | -5.00±0.71 | -5.21±0.70 |
| Trend after 2001 | 1.91±0.94 | 2.26±1.04 | 2.00±1.04 |
| $\chi$ | 10.61 | 11.82 | 11.71 |
| | Only with HF and PWT | | |
| R$^2$ | 0.84 | 0.77 | 0.82 |
| Trend before 2001 | -5.34±0.84 | -4.79±0.97 | -5.23±0.89 |
| Trend after 2001 | 2.04±1.24 | 2.83±1.48 | 2.13±1.31 |
| $\chi$ | 14.04 | 16.03 | 14.81 |

**Table 3: Idem Table 2 for Sept15-Oct15 period. SAT dataset is also presented.**

| | Multi-Sensor Reanalysis (MSR-2) | | | Composite satellite data (SAT) | | |
|---|---|---|---|---|---|---|
| | 400K-600K | 475K | 550K | 400K-600K | 475K | 550K |
| $R^2$ | 0.95 | 0.94 | 0.94 | 0.96 | 0.94 | 0.94 |
| Trend before 2001 | -5.81±0.60 | -5.55±0.66 | -5.63±0.77 | -5.86±0.57 | -5.57±0.64 | -5.64±0.65 |
| Trend after 2001 | 1.42±0.92 | 1.73±1.01 | 1.58±1.02 | 1.70±0.87 | 1.96±0.99 | 1.79±0.99 |
| $\chi$ | 9.67 | 10.65 | 10.77 | 9.21 | 10.39 | 10.46 |
| | Only with GRAD, HF and PWT | | | | | |
| $R^2$ | 0.94 | 0.93 | 0.93 | 0.95 | 0.93 | 0.93 |
| Trend before 2001 | -5.86±0.56 | -5.71±0.64 | -5.67±0.64 | -5.93±0.56 | -5.75±0.66 | -5.70±0.63 |
| Trend after 2001 | 1.21±0.83 | 1.42±0.95 | 1.35±0.94 | 1.40±0.83 | 1.56±0.97 | 1.47±0.93 |
| $\chi$ | 9.35 | 10.68 | 10.65 | 9.35 | 10.84 | 10.55 |
| | Only with HF and PWT | | | | | |
| $R^2$ | 0.87 | 0.82 | 0.86 | 0.88 | 0.83 | 0.87 |
| Trend before 2001 | -5.89±0.84 | -5.74±0.98 | -5.70±0.86 | -5.96±0.82 | -5.78±1.00 | -5.72±0.84 |
| Trend after 2001 | 1.45±1.24 | 1.70±1.45 | 1.57±1.27 | 1.63±1.21 | 1.82±1.47 | 1.68±1.24 |
| $\chi$ | 14.06 | 16.40 | 14.39 | 13.71 | 16.18 | 14.03 |

Table 4: Coefficient of determination $R^2$, trends $\pm 2\sigma$ in Mt yr$^{-1}$ before and after the turnaround year 2001 derived from multi-regression model using OMD dataset (MSR-2 total ozone columns and threshold of 220 DU, see the text) for September and 15Sept-15Oct over 1980-2017 period. The residual is represented in DU by $\chi$ as explained in Tab. 2.

|  | September | 15Sept-15Oct |
| --- | --- | --- |
| $R^2$ | 0.85 | 0.91 |
| Trend before 2001 | 1.28±0.25 | 1.59±0.24 |
| Trend after 2001 | -0.78±0.39 | -0.68±0.37 |
| $\chi$ | 4.04 | 3.85 |
| Only GRAD, HF and PWT | | |
| $R^2$ | 0.82 | 0.90 |
| Trend before 2001 | 1.29±0.24 | 1.61±0.22 |
| Trend after 2001 | -0.86±0.36 | -0.65±0.33 |
| $\chi$ | 4.04 | 3.68 |
| Only HF and PWT | | |
| $R^2$ | 0.78 | 0.85 |
| Trend before 2001 | 1.29±0.26 | 1.61±0.27 |
| Trend after 2001 | -0.88±0.38 | -0.70±0.39 |
| $\chi$ | 4.37 | 4.44 |

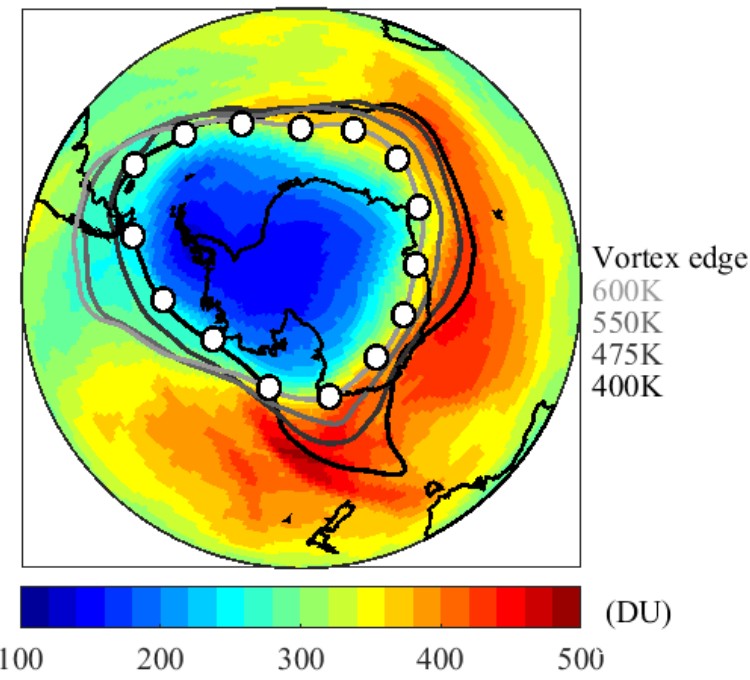

**Figure 1: Total ozone (DU) from MSR-2 on October 7, 2012 at 12 UT. Vortex edge position at different isentropic levels are auditioned to the map and represented by black to light grey lines. White dot marks identify the region considered inside the vortex using the 400 K -600 K range classification.**

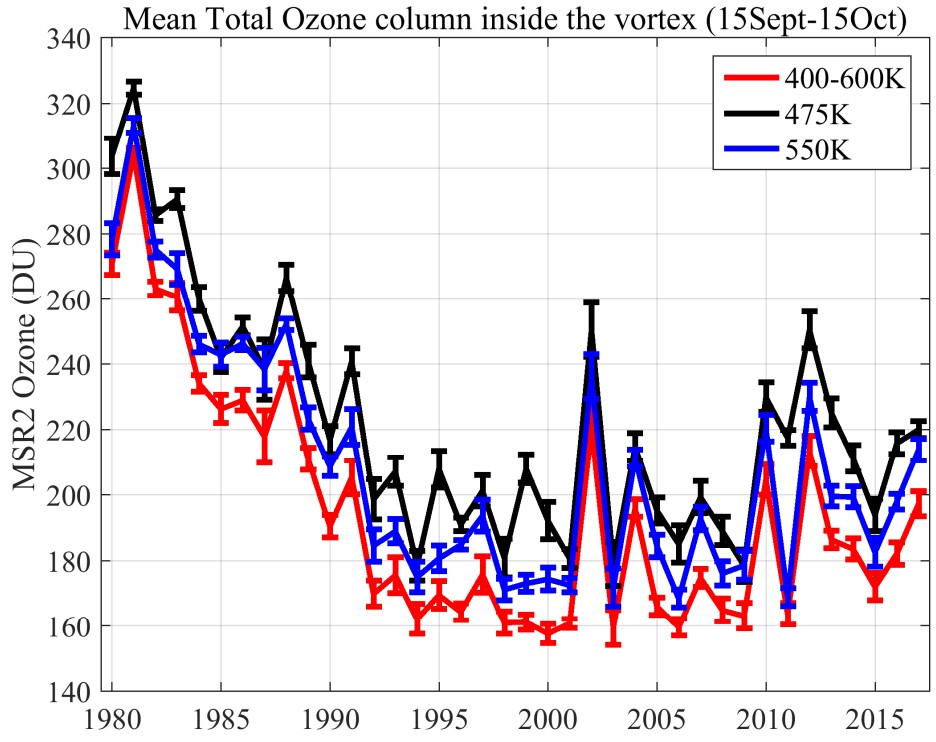

**Figure 2:** Evolution of total ozone of MSR-2 dataset inside the vortex averaged each year on 15Sept–15Oct period for different classifications: standard method at 475 K and 550 K represented by black and blue lines, respectively and method considering the 400 K-600 K altitude range by red line. Error bars represent twice the standard error.

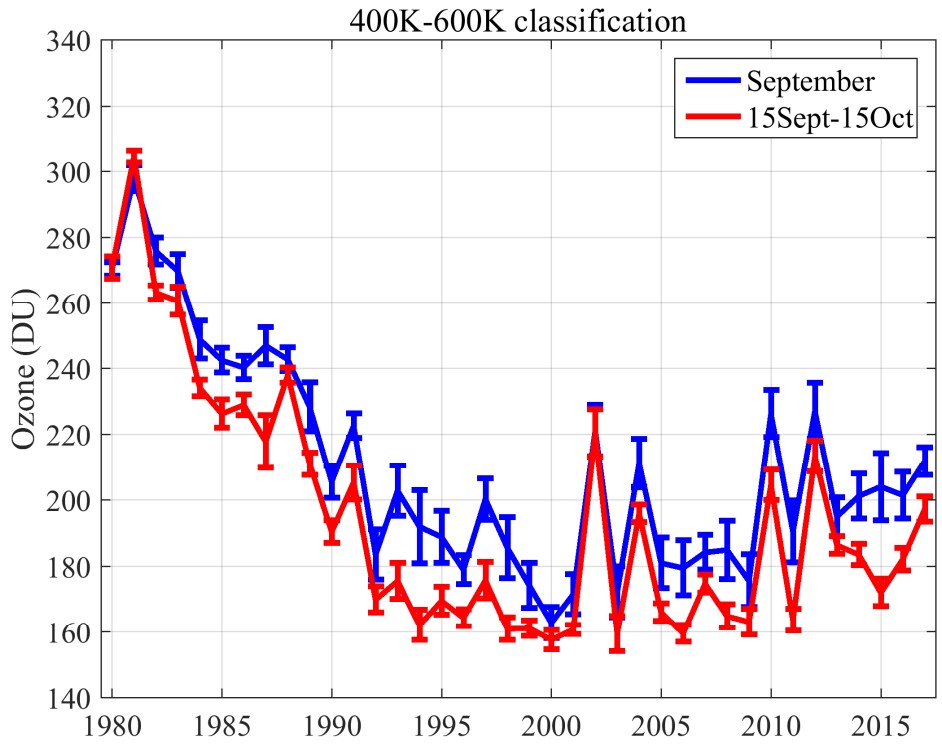

**Figure 3: As in Fig. 2 but only for 400K-600K classification on different periods: September and mid-September to mid-October. Error bars represent 2σ.**

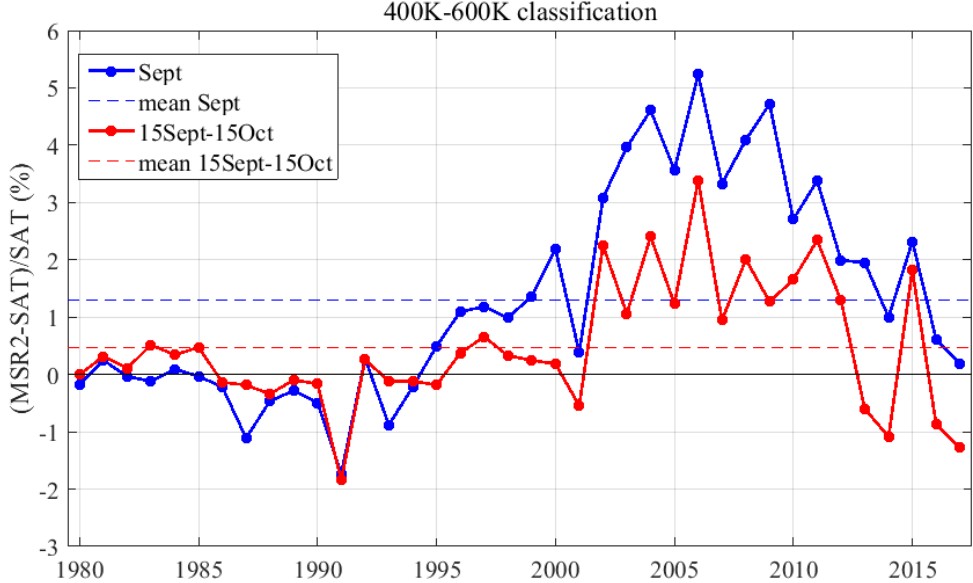

**Figure 4: Relative difference between MSR2 and SAT mean total ozone inside the vortex for September (blue curve) and 15Sept-15Oct (red curve) periods. Horizontal dash lines correspond to the mean bias between data series.**

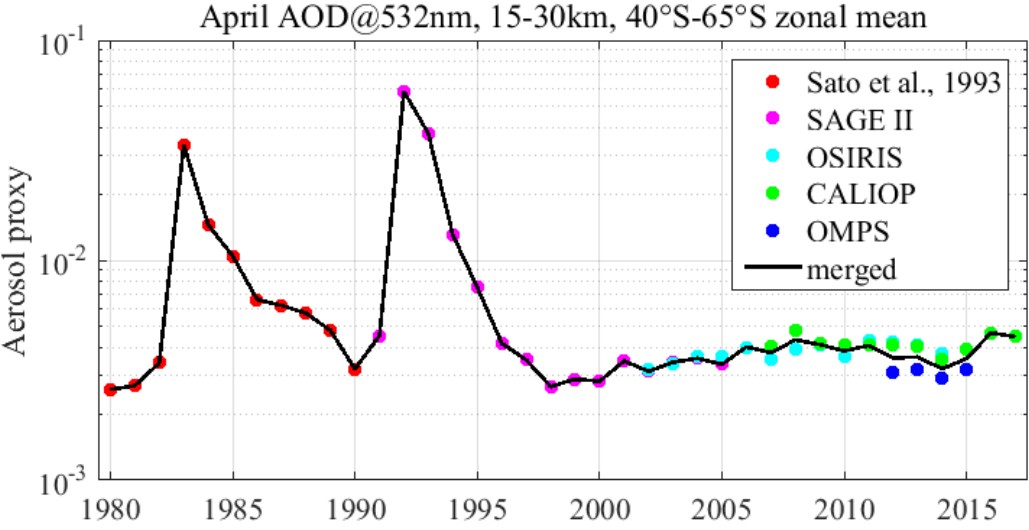

**Figure 5: Time series of April monthly mean AOD at 532 nm within 40°S-65°S and 15-30 km of normalised Sato et al., 1993 dataset (see main text) and from satellites (SAGE II, OSIRIS, CALIOP, OMPS). The corresponding merged data is represented by the bold line.**

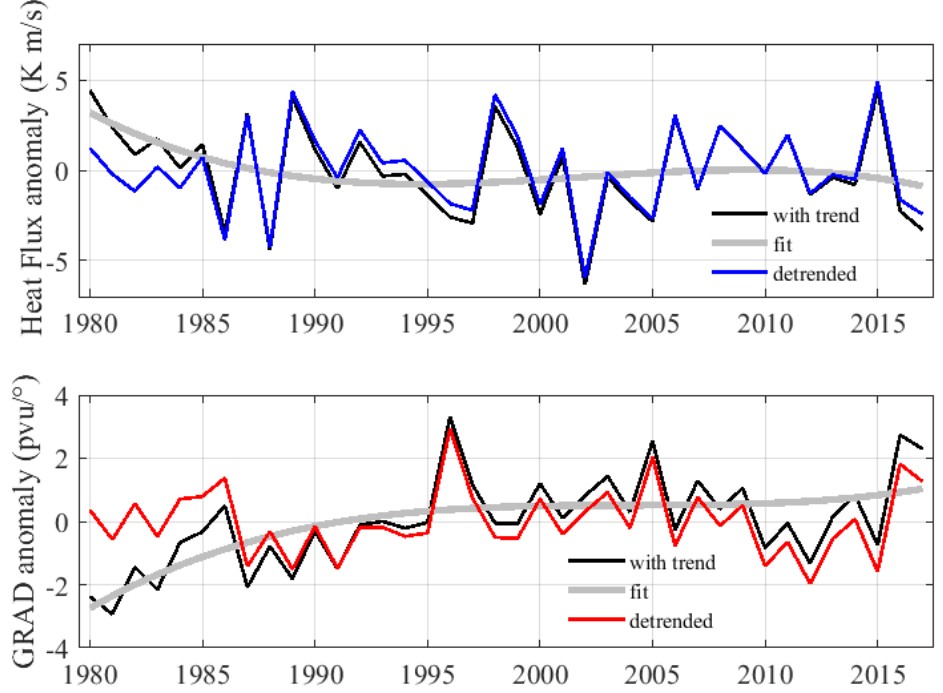

**Figure 6: Heat Flux (top panel) and Gradient - GRAD (bottom panel) anomalies for the 15Sept-15Oct period: before removing a polynomial fit of 3rd order (black line), fit (grey line) and after removing the fit (blue/red line).**

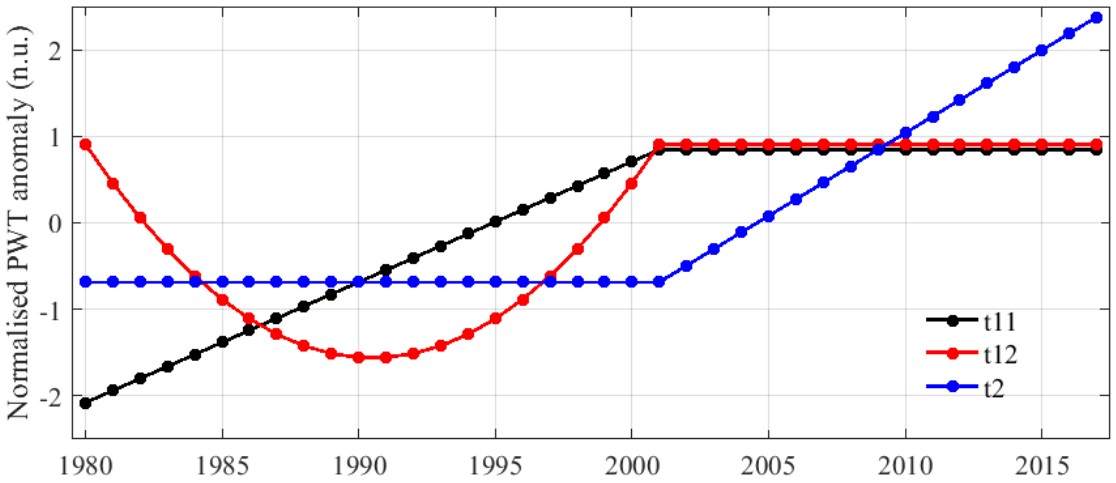

**Figure 7: Anomalies of the linear functions before and after 2001 (t11 and t2, respectively) and parabolic function (t12) that correspond to the PWT proxy (see Eq. 2 to 5). Each proxy anomaly is normalised by the corresponding standard deviation.**

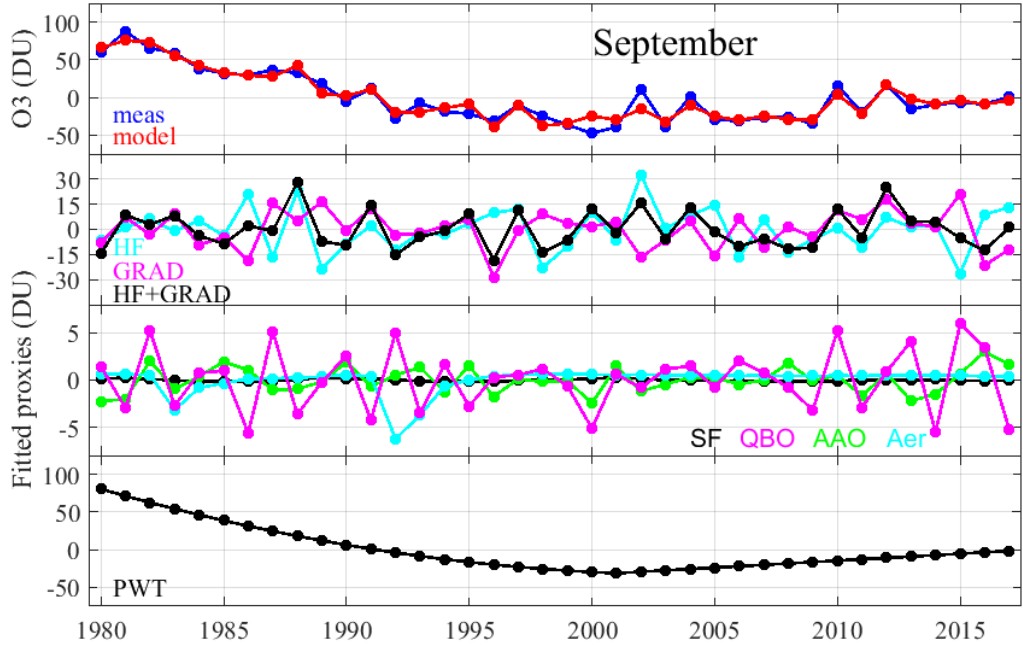

**Figure 8: Deseasonalised total ozone inside the vortex of MSR-2 series (meas) and regression model (model) for September using 400 K-600 K classification (top panel). Contributions of proxies are also shown: Heat Flux - HF, gradient - GRAD and the combination of both HF+GRAD (second panel); solar flux - SF, QBO (QBO at 30hPa + QBO at 50hPa), Antarctic Oscillation - AAO and Aerosol - Aer (third panel); and PWT (bottom panel). Ozone anomalies and contributions of proxies are given in DU.**

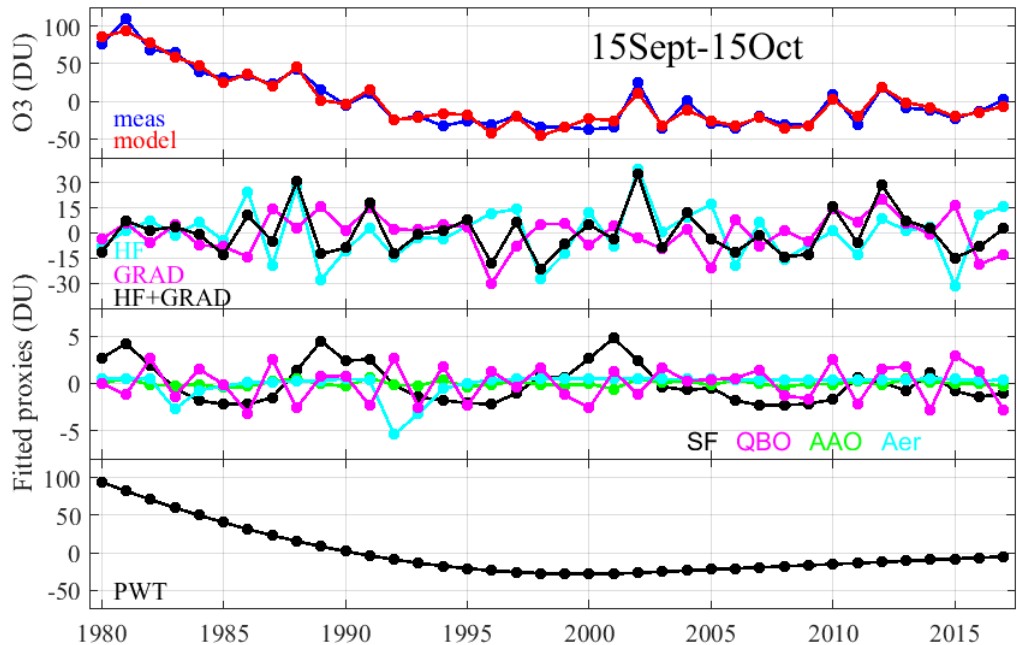

**Figure 9: As in Fig. 7 for 15Sept-15Oct.**

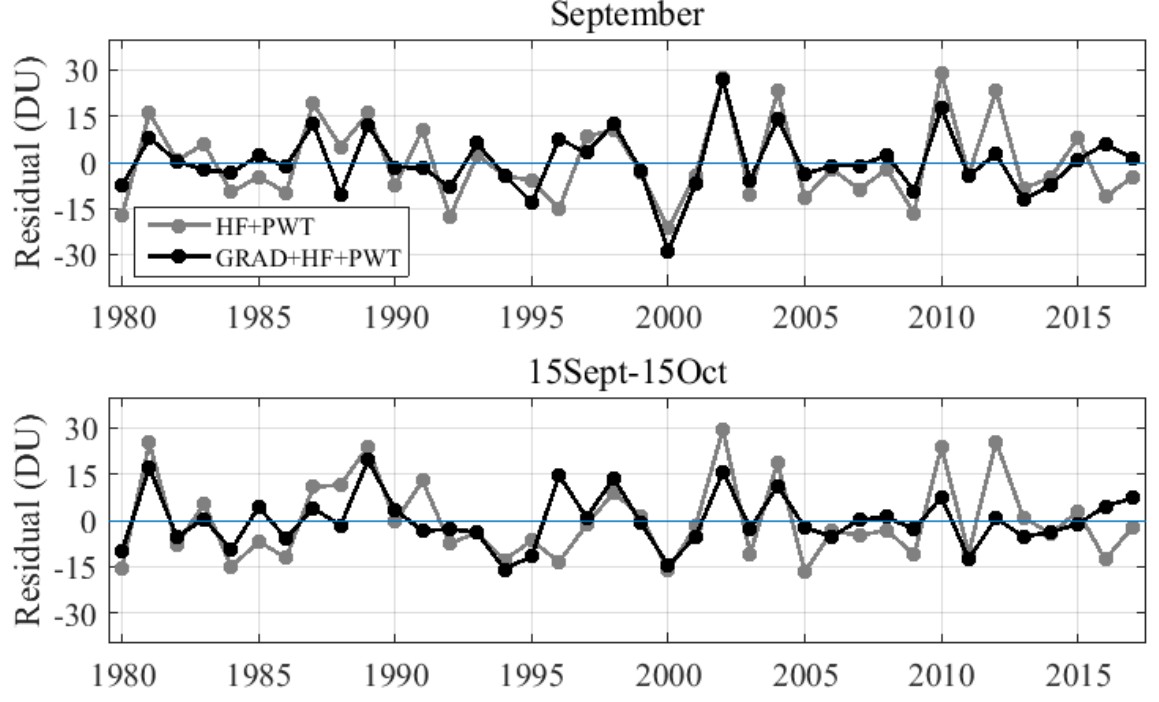

**Figure 10: Residual (in DU) with and without contribution of GRAD proxy for September (top panel) and 15Sept-15Oct period (bottom panel)**

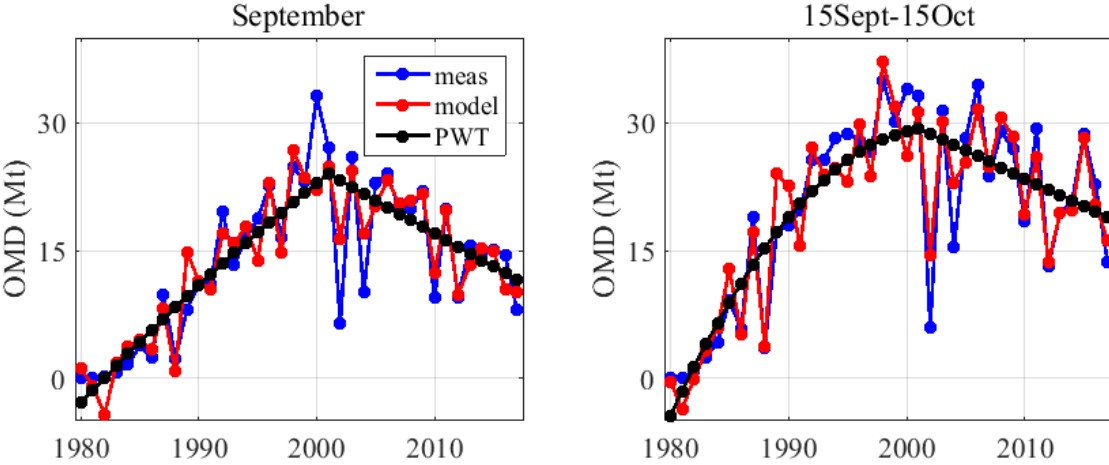

**Figure 11: OMD (in Mt) computed from total columns of MSR-2 dataset lower than 220 DU and south of 60°S for September (left panel) and 15Sept-15Oct (right panel). Regressed values by MLR analysis using GRAD, HF and PWT are also shown as well as the fitted PWT proxy.**

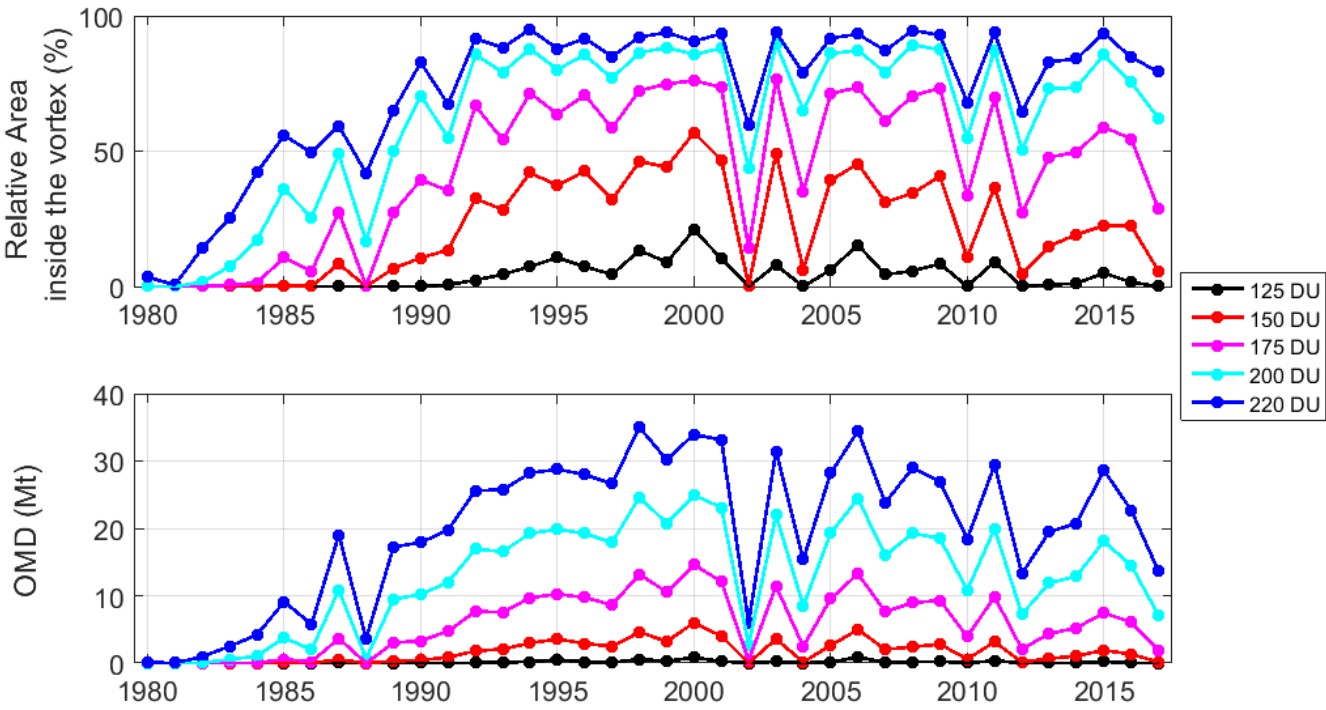

**Figure 12: Relative area inside the vortex (in %) with values lower than 5 level thresholds (125, 150, 175, 200 and 220 DU) computed from MSR-2 dataset using 400 K-600 K classification on 15Sept-15Oct period (top panel). OMD (in Mt) time series computed from MSR-2 total ozone data for the same 5 thresholds and time period are displayed in the bottom panel.**

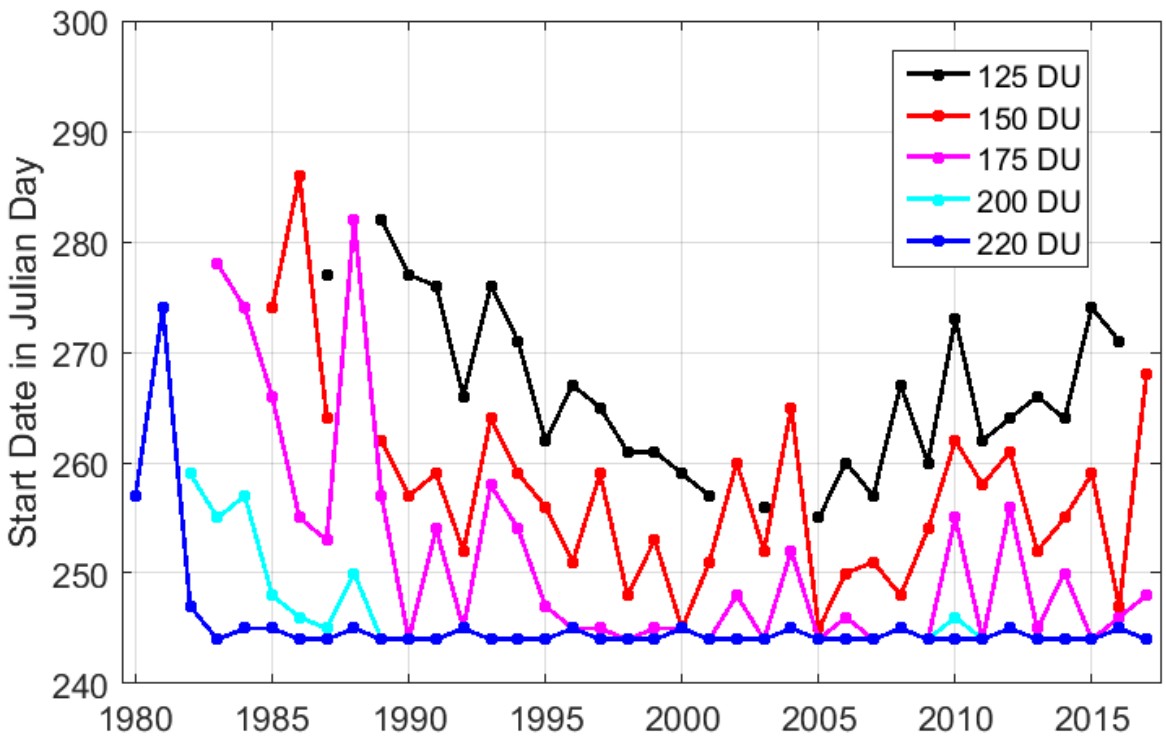

**Figure 13: Start day of occurrences of total ozone levels lower than different thresholds (125, 150, 175, 200 and 220 DU) computed from the MSR-2 dataset using the 400 K-600 K classification between September 1st and October 15th.**