# Peer review of "Multiple symptoms of total ozone recovery inside the Antarctic vortex during Austral spring"

_Atmospheric Chemistry and Physics, 2017_

## Referee Comment (RC1) · A. de Laat (Referee) · 29 Dec 2017

Review of ACP(D) paper 10.5194/acp_2017-1157

"Symptoms of total recovery inside the Antarctic vortex during Austral spring"

by Pazmino et al. [2017]

This paper discusses recovery of springtime stratospheric ozone within the Antarctic stratospheric polar vortex based on total ozone column measurements. Or recovery of the so-called Ozone Hole.

The method applied is a multi-variate regression, used to estimate and separate the influence of various atmospheric processes that are known to affect Antarctic springtime stratospheric ozone depletion.

This is a well-known and widely used methodology in previous papers looking into this topic. This paper thus also builds on previous work and publications on the topic that have appeared in scientific literature especially in the last decade.

As outlined below, I think this paper is publishable but some important revisions are needed.

There is one recommendation [1] that the authors I hope are willing to consider but would be a considerable amount of work, and which I could live with if not part of the revision given that suggestions for revision [2] are incorporated.

Full review

Because of the legacy of the topic, it appears useful to consider what is new in this paper compared to what has already been published.

- Longer time period (1979 – 2015)
- A new proxy for the regression model
- A new/different method to estimate the vortex edge (needed for calculation of the annual average amount of springtime Antarctic ozone depletion)
- An piece-wise time trend based on a combination of a linear function and a polygon (second order; quadratic)
- Discussion of results for two periods (September average and 15 September – 15 October average). The latter is not a commonly used time period. The choices made in the paper will be discussed later on.
- Analysis of alternative Antarctic Ozone Hole metrics (area with total ozone columns < 150 DU and < 125 DU as compared to the standard 220 DU Ozone Hole area).

It is also useful to consider what is more or less new with regard to the findings of the paper

- Most proxies used in the regression do not reduce trend uncertainties. Piece-wise trends and heat fluxes alone (with or without the new GRADS proxy) explain more than 90% of the long term variability. Hence, based on this paper it could be argued that most proxies could be discarded, which is consistent with previous work.
- The longer time period considered leads to higher statistical significances of the post-peak trends in Antarctic springtime stratospheric ozone (from 2001 onwards; as expected based on previous papers).
- Higher statistically significant trends for the September period compared to 15 Sep – 15 Oct (consistent with previous findings)

Major comment.

This paper relies on a limited set of ozone records (Sep average, 15 Sep – 15 Oct average; area 220/150/125 DU), and a limited set of proxies used in the multi-variate regression. In two recent papers [de Laat et al., 2015, 2017; 2016JD025723], we explore the uncertainty ranges associated with the choices that can be made with regard to the time period over which the ozone parameter is calculated, and uncertainties associated with proxies as used in multi-variate regressions.

A paper like this, and also most previous papers on the subject, thereby only consider a few options in a much larger parameter space of options. This has the risk that it limits the view and interpretation (tunnel vision). The few time series that are looked at are then seen as the truth, every wiggle becomes meaningful, and too much attention is given to the formal statistical significances, whereas structural uncertainties are important as well.

For example, we have shown that rather arbitrary choices with regard to the proxies used in the regression have a strong impact on the formal statistical trend errors. We therefore argued that structural uncertainties are much larger than the formal statistical trend errors, which is important for confident statements about whether recovery has started or not. The same applies for the time period over which the ozone metric of choice is calculated. We see considerable differences in trends and trend uncertainties.

Furthermore, we also argued in de Laat et al. [2017] for the use of the ozone mass deficit rather than average ozone or area as the preferred metric to study long term changes in springtime Antarctic stratospheric ozone depletion. The motivation was that the OMD suffers less from what is discussed above (arbitrary choices) then average ozone and area.

This paper does not address these issues, nor are results put in the context of this work. The paper does show and confirm that most proxies in these multivariate regressions are not really useful. Confirms that statistical significances of post-peak trends become better because of a longer record (but which has to, given the mathematical nature of linear regressions). Confirms that there are differences in trends between September and 15 Sep – 15 Oct. And confirms that there are uncertainties associated with several parameters that need to be defined in advance (vortex position, vortex stability).

But there is no real discussion about why these are the appropriate choices. The GRADS proxy helps in improving the explained variability. But is that the justification? Smaller residuals? If so, I'm sure even better proxies can be constructed. Furthermore, the GRADS proxy is detrended. Why? If the GRADS proxy truly represents a physical process, why isn't GRADS allowed to also change on longer timescales (note that this is a point of contention in recent literature: is recovery fully attributable to ODSs or are there other long term changes in atmospheric dynamics that also play a role?). The same is true for use of the parabolic trend. It is not the standard approach in regression studies (all studies cited use PWLT), but the effect using two linear trends or one parabolic and one linear trends is not discussed (as far as I could see). It could also be argued based on figure 1 that none of the vortex edge definitions really captures only the vortex core. All still capture some high ozone columns around the vortex edge, which likely introduces variability in the ozone record not related to inner-vortex ozone depletion.

Consider that the standard 220 DU value used for OMD and even area fall well inside the 600 K vortex edge.

This is an exhaustive list of issues, which is exactly the point we want to make here: the issues raised in recent literature about arbitrariness of choices that are made, and the corresponding risk of tunnel vision.

Note that this is also why in de Laat et al. [2017] it is proposed to step away from the whole regression business.

This paper does show that ozone variability is mostly governed by depletion (ODSs) and heat fluxes or vortex (in)stability. How to properly account for the heat fluxes or vortex (in)stability is, however, not really clear, and this paper introduces yet another approach. In de Laat et al. [2017] it is instead proposed to simply remove the years that are characterized by a more unstable vortex from the record. Such years can be easily identified, but how they affect ozone depletion is much more complex, and appears to depend for example on the exact timing of vortex disturbances [de Laat and van Weele, 2011; 10.1038/srep00038]. This paper provides some additional ammunition for the proposal to step away from the regression methods.

The presence of this exhaustive list of issues and questions would be less of a problem if the paper introduced new concepts or new ideas, but the paper mostly builds on previous work and confirms what other papers have also concluded.

The new concepts and ideas that are introduced in the paper do not help in clarifying in what has recently emerged: the sometimes large structural uncertainties in this particular field of research, and arbitrariness with which analyses are performed. If anything, they only confirm the existence of large structural uncertainties and the arbitrariness.

So, how do I think this paper could be improved?

[1] One possibility would be to include additional analyses cover more of the parameter space. The paper already also uses average ozone and area, so a mass deficit could be included as well (see Fig. 5 of de Laat et al. [2017]).

The use of different area definitions based on different ozone thresholds could also be expanded – like looking at changes in the probability distributions of total ozone (a bit like Yang et al. [2008; 10.1029/2007JD009675], but much more extensive). However, that would require a considerably amount of additional work.

I could live without such an analysis if:

[2] regardless, results should be discussed within the context of recent publications and criticism of existing methods of Antarctic stratospheric ozone recovery detection. This is currently lacking, as also reflected in the conclusions section, which is more of a summary than a conclusion.

The challenge here will be to discuss it in such a way that that discussion does not undermine the findings of the paper.

So, what should be discussed are what I consider the most important findings of the paper:

- Most proxies in the MVR do not contribute much (if anything) to reduce trend uncertainties (small explanatory power)
- September yields a higher statistical trend significance than 15 Sep-15 Oct.
- Range of trend values and trend significance levels are indicative (or not) for structural uncertainties and systematic errors (this needs to be further supported)

In addition, I think the following should be included in a revision:

- report 2000-2010 and 2000-2012 trends & statistics for comparison with the 2000-2015 trends (and significances). This is helpful for comparison with results from older previous papers using MVR methods but somewhat different proxies.

- Use of "area" for 150 DU of 125 DU is an interesting  more or less novel approach. Results show that such small TCOs did not occur until the late 1980sand early 1990s, indicative that these parameters are more sensitive for more severe ozone depletion. This also means that these parameters should return back to zero values earlier than the TOC columns return to 1980 levels. This method/analysis could be expanded more, by using the 150 or 125 DU also as vortex edge proxies (average ozone within area), and for Ozone Mass Deficit calculations (which traditionally is based on the 220 DU level but that is somewhat arbitrary). Possibly also report 175 and 200 DU results.

   (in all honesty, I think the analysis of long term changes in probability distributions could be a topic of a completely separate paper)

Minor comments

Page 1, line 25-26, and line 29 (and correspondingly tables 2 & 3), in particular the range of trend values that are reported.

How should this range be interpreted? Could this be considered representative of the structural uncertainty?

Page 2, line 14-15, the explanation of why October ozone behaves differently from September ozone.

October ozone is partly governed by different processes than September ozone. First of all, catalytic photochemical ozone destruction ceases in October. Rather, there is regeneration of ozone due to photolysis of O2 and oxidation of CH4 and carbon monoxide [Grooss et al., 2011; 10.5194/acp-11-12217-2011]. Furthermore, there is continued downward transport of ozone rich outer-vortex air into the vortex from the upper stratosphere down to the lower stratosphere [de Laat and van Weele, 2011; doi:10.1038/srep00038]. And there is vortex dynamics, as the authors correctly note. Together, these processes to a large extent determine October Antarctic inner vortex ozone.

Page 4, line 15. It is stated that a 5-day smoothing is applied to the EL of the maximum PV gradient. However, as far I know Nash et al. [1996] does not call for a 5-day smoothing. If that is right, then what is the justification of the 5-day smoothing?

Page 5, line 6-7. Correlations. Sometimes the paper uses R, sometimes $R^2$. Be consistent, preferably using $R^2$ and only refer to R if the correlation is negative (still providing $R^2$).
See also: page 5 - line 16, Page 10, line 17, and make sure to check throughout the paper.

Page 5, figure 4. The differences between SAT and MSR2 are fairly straight forward to explain. Up until 1993, both rely solely on TOMS. From 1993-1995, MSR2 relies on SBUV, and thanks to the data assimilation gaps are filled. From 1996 onwards, MSR-2 also uses GOME (1996 to 2005), SCIAMACHY (2002-2012), OMI (2004-), and GOME2 (2007-). Furthermore, MSR-2 uses ground-based total column data to account for inter-instrument differences. As a result, the estimated average MSR-2 total ozone column bias has been estimated at 1% [van der A et al., 2015; amt-8-3021-2015].

Add to line 21 the following "whole vortex. The data assimilation of MSR2 to some extent does fill gaps when ozone measurements are limited."

Add after line 25. These differences are caused by MSR-2 starting to use multiple satellite total ozone column records after 1996, the procedures in MSR-2 to account for inter-instrument differences, and the data assimilation methodology that allows for filling gaps [van der A et al., 2015].

Page 6, line 25. It is stated that both PWLT and a combined parabolic trend – linear trend is generally used. The latter is not true, all papers cited only rely on a PWLT. The parabolic trend is a new concept introduced in this paper. As such, it should be explained later in the paper what

the differences are that associated with both PWTs (the PWLT appear no to be used in the paper at all).

Grammar, typos.

Page 1, line 27. Replace "lower than" with "smaller than"
Page 2, line 4: change to "interannual variability of ozone as a function of the 11 year"
Page 2, line 8. I assume what is meant is "for the period over which the ozone record is calculated and for …"
Page 2, line 12. "ozone content is deepest", I think what is meant here is "where ozone depletion is largest" or "where the ozone deficit is largest".
Page 2, line 19. "update of the ozone"
Page 2, line 22. "full development of Polar ozone depletion". I think what is meant here is "the period of fastest catalytic photochemical ozone destruction"
Page 3, line 35. Include reference to de Laat et al. [2017; 10.1002/2016JD025723] as a paper that also uses MSR2.
Page 4, line 15. Change to "This limit is subsequently smoothed temporally with"
Page 4, line 17. Start with "The Nash criterion"
Page 4, line 29. Change to "On this particular day, the region …"
Page 4, line 32. Change to "consist of"
Page 4, line 35. Change to "using the new classification."
Page 4, line 36. Change to "The standard classification estimates a 40 DU and 20 DU larger ozone mean …"
Page 5, line 3. Change to "for the SAT data series … … based on the single …"
Page 5, line 4. Change to "Error bars represent the two sigma …"
Page 5, line 7. Change to "at the 2σ level"
Page 5, line 12. Change to "is preferred since it takes …"
Page 5, line 35. Change to "The ODS contribution to long-term trends in ozone is represented by piece-wise linear trend …"
Page 6, line 15. Start new paragraph after "period"
Page 6, line 21. Change to "with a p-value"
Page 7, line 14-17. Rephrase line "Despite … Weber et al. 2017)". I assume you want to note that although September shows large variability in total ozone, it is still a commonly used month for recovery detection.
Page 7, line 18. Remove "are highlighted", change "conclude that" to "identify "
Page 7, line 18. Change "on October" to "for October"
Page 7, line 20-21. Delete "In our study … previous section."
Page 7, line 25. Change to "the year 2000 was characterized by …"
Page 7, line 26. Change to "September, and yields a relatively high …"
Page 8, line 7. Add reference to Chipperfield et al. [2017; doi:10.1038/nature23681], who amongst others discuss the differences in pre-post peak ozone recovery rates.
Page 9, line 30. Change to "at 550K where the trend after …"
Page 9, line 37-38. Change to "Trends estimate for the second period show slightly"
Page 9, lines 40-41. Please rephrase, I don't fully understand what is meant here.
Page 10, line 1. Change to "higher than 3, the threshold value …"
Page 10, line 21. Change to " The ozone hole is also frequently defined as …"
Captions of figure 11 + 12: OMIT → OMI

---

## Referee Comment (RC2) · Anonymous Referee #2 · 5 Feb 2018

The goals of this paper are described in the introduction: "...to provide an update of ozone evolution inside the Antarctic vortex during the last decades taking into account the vortex baroclinicity. The main aim is to determine the different contributions to ozone interannual variability and to estimate the post 2001 total ozone trend and related significance for different periods: September...and mid-September to mid-October when the maximum ozone loss is reached."

I concur with much of what the other reviewer articulated, in particular these points from DeLaat's review:

1. "The presence of this exhaustive list of issues and questions would be less of a problem if the paper introduced new concepts or new ideas, but the paper mostly builds on previous work and confirms what other papers have also concluded."

[Figure]

2. "This paper does not address these issues, nor are results put in the context of this work."

3. "The few time series that are looked at are then seen as the truth, every wiggle becomes meaningful, and too much attention is given to the formal statistical significances, whereas structural uncertainties are important as well. For example, we have shown that rather arbitrary choices with regard to the proxies used in the regression have a strong impact on the formal statistical trend errors. We therefore argued that structural uncertainties are much larger than the formal statistical trend errors, which is important for confident statements about whether recovery has started or not."

I especially agree with DeLaat's concerns about the 'structural uncertainties' in this regression analysis, so please address all issues described in his review. In addition, there are other issues below related to ozone data sets that need to be addressed in a revised manuscript. If revisions are made that address both DeLaat's and my review, this paper could be published in ACP.

Specific topics of Concern

The composite satellite total ozone time series, referred to as SAT. The merging of satellite data sets into a single record is something to be done very carefully. Instrument measurements have bias and drift, and combining data sets in order to extract small trends (i.e., ozone recovery) requires a great deal of care and a good deal of knowledge about each instrument's characteristics and sampling pattern (i.e., coverage). I see no evidence here that any such considerations were used when combining the data sets. In fact in Figure 4, the difference between the assimilated ozone time series (MSR) and the SAT shows big jumps! There is a large trend from 1990-2005. Does this represent an unphysical trend (i.e., changes in the observing system) in the assimilation, or is this coming from how the individual data sets in the SAT were merged? Have you tried your trend analyses on the 5 merged ozone data sets referenced in Weber et al. [2017]? Without any discussion or justification of how the data sets were merged in this study,

[Figure]
* * *
Interactive
comment

I don't see how the trend results presented here (and especially their uncertainties!) can be taken seriously.

The 'range method' is not clearly explained. I understand that you are using it to see the sensitivity of the calculated trends to the definition used for the area of depletion, and I get that you calculate different areas depending on which isentropic level is used, but exactly how are you deciding which levels to use? Are you averaging over all the 400-600K level results? Only some of them? Do you choose the same range for each year? The details of this methodology were not made clear. It's interesting that in the end you conclude that the 475K results are as good as the other definitions. Is this because this is an altitude where there is some of the most severe depletion? An explanation for this result should be offered.

The satellite instruments used (all UV sensors) do not see to the south pole in early September. The analysis calculated results for the polar region for the entire month of September, but measurements cannot be made at the highest latitudes in early September. Thus the 'September average' will be more strongly weighted by lower latitudes and later September dates. Please describe how the satellites' sampling of the polar area varies over September and what this does to the 'September averaged' quantity. This may impact the meaning of the trend results as they will include more of the late September, higher dynamical variability measurements.

---

## Author Comment (AC1) · 12 Apr 2018

The comment was uploaded in the form of a supplement:
https://www.atmos-chem-phys-discuss.net/acp-2017-1157/acp-2017-1157-AC1-supplement.zip

---

## Author Response (AR1)

**Reply to A. T. J. de Laat review of manuscript acp-2017-1157**

**Symptoms of total ozone recovery inside the Antarctic vortex during Austral spring**

Andrea Pazmino on behalf of all co-authors

We thank A. T. J. de Laat for the important and helpful review of our manuscript. Many interesting suggestions were incorporated to the new version of the manuscript. Please find our answers (in red) in three different sections: Comments to full review (1), Reply to major comments (2) and Reply to minor comments (3)

**1. Comments to full review**

Because of the legacy of the topic, it appears useful to consider what is new in this paper compared to what has already been published.
- Longer time period (1979 – 2015)
- A new proxy for the regression model
- A new/different method to estimate the vortex edge (needed for calculation of the annual average amount of springtime Antarctic ozone depletion)
- An piece-wise time trend based on a combination of a linear function and a polygon (second order; quadratic)
- Discussion of results for two periods (September average and 15 September – 15 October average). The latter is not a commonly used time period. The choices made in the paper will be discussed later on.
- Analysis of alternative Antarctic Ozone Hole metrics (area with total ozone columns < 150 DU and < 125 DU as compared to the standard 220 DU Ozone Hole area).

It is also useful to consider what is more or less new with regard to the findings of the paper

- Most proxies used in the regression do not reduce trend uncertainties. Piece-wise trends and heat fluxes alone (with or without the new GRADS proxy) explain more than 90% of the long term variability. Hence, based on this paper it could be argued that most proxies could be discarded, which is consistent with previous work.
- The longer time period considered leads to higher statistical significances of the post-peak trends in Antarctic springtime stratospheric ozone (from 2001 onwards; as expected based on previous papers).
- Higher statistically significant trends for the September period compared to 15 Sep – 15 Oct (consistent with previous findings)

We appreciate your time and your general comment about our work, which allowed us to improve the paper. Since total ozone data is now available for MSR-2 until the end of October 2017, we decided to extend our study to the year 2017 using SAT and MSR-2 data. Due to this extension, all figures of the manuscript have been revised, except Figure 1 where white dot marks where added to highlight the region considered inside the vortex by the 400 K-600 K classification range. In addition we have noticed that the figure 12 of the original manuscript about the time shift of low values was not very clear. A new figure, Figure 13, has been produced in order to better illustrate the time delay in appearance of low total ozone values within the vortex. Similar conclusions as in the original version of the manuscript were provided. Furthermore the word "Multiple" was added to the title to highlight the fact that different signs of recovery were obtained in this work, e.g. (1) Significant positive trends of total ozone since 2001 in September and for the first time in the period of maximum ozone depletion (15Sept-15Oct) using MLR analysis on average ozone inside the vortex and Ozone Mass Deficit, (2) Decrease of occurrences of very low ozone values within the vortex and (3) increased delay of occurrence of low total ozone levels in the September 1st – October 15th period.

We generally agree with your appreciation of what is new and what is less new in our paper. Regarding the former, as you mention, one of the novelty of this work is to consider several different isentropic levels in the range 400 K – 600 K to make the classification based on the well-known Nash Criterion, in order to better constrain the ozone value inside the vortex. We think also that the addition of the GRAD proxy, based on physical considerations, provides a better agreement between observation and regressed values. The study of the very low ozone values within the vortex, based on different thresholds, provides also interesting indices towards ozone recovery.

We agree that it is also important to highlight the results confirming previous ones. Some of recent works using MLR have been already mentioned in the paper (Chipperfield et al., 2017; Weber et al, 2017) and also using other methods (Solomon et al., 2016).

It is true that most proxies in our MLR analysis do not significantly reduce trend uncertainties and piece-wise trends added to heat flux can explain more than 80% of vortex variance, but it is interesting to evaluate the contribution of proxies that are commonly used.

Longer time series generally results in higher statistical significance but due to higher ozone interannual variability in the last decade, each year can count in the trend analysis, considering the still relatively short ozone records since 2001.

**2. Reply to Major comments**

This paper relies on a limited set of ozone records (Sep average, 15 Sep – 15 Oct average; area 220/150/125 DU), and a limited set of proxies used in the multi-variate regression. In two recent papers [de Laat et al., 2015, 2017; 2016JD025723], we explore the uncertainty ranges associated with the choices that can be made with regard to the time period over which the ozone parameter is calculated, and uncertainties associated with proxies as used in multi-variate regressions.

Our work builds on previous studies and especially on recommendations made in de Laat et al. (2015) to optimize the multi-linear regressions. One of the purposes of this paper is to reproduce the variation of ozone inside the vortex during the last decade, especially from 2010 where increased variability is observed. This is how we came up with the GRAD proxy related to vortex stability during the studied month/period. This proxy is linked to the potentially mixing between inside and outside vortex regions during the period.

Further, in order to take into account the rounding off of the ozone loss due to saturation since the 1990s, which is especially visible by the end of September/beginning of October, we included a polynomial function to the linear functions used to evaluate long-term trends.

A paper like this, and also most previous papers on the subject, thereby only consider a few options in a much larger parameter space of options. This has the risk that it limits the view and interpretation (tunnel vision). The few time series that are looked at are then seen as the truth, every wiggle becomes meaningful, and too much attention is given to the formal statistical significances, whereas structural uncertainties are important as well.

For example, we have shown that rather arbitrary choices with regard to the proxies used in the regression have a strong impact on the formal statistical trend errors. We therefore argued that structural uncertainties are much larger than the formal statistical trend errors, which is important for confident statements about whether recovery has started or not. The same applies for the time period over which the ozone metric of choice is calculated. We see considerable differences in trends and trend uncertainties.

We have considered different scenarios (2 ozone datasets, 3 different proxies' combinations, different criteria for vortex limit). Our MLR analysis could reproduce very well ozone in the last decade and we show that the GRAD proxy, based on physical explanation, improves the agreement between observation and regressed values. A robust estimation of structural uncertainties requests a "big-data" treatment as in de Laat et al. (2015). This was done already and is out of the scope of our study. However, a comparison of the maximum trend difference between the scenarios considered in the study and the retrieved trend uncertainties provides some evaluation of the structural uncertainty of our analysis.

The following paragraph was added in the conclusions of the marked-up version of the paper (page 14, line 26 to line 33):

*"The structural uncertainties of the MLR analysis linked to the selection of proxies were not fully analysed in this work, as in De Laat et al. (2015). The main sensitivity tests concerned the baroclinicity of the vortex and the impact of its stability during the studied periods. Trend differences in the various scenarios analysed provide some quantification of related uncertainties and are lower than the statistical trend uncertainties. Further, the large determination coefficients obtained for both periods analysed give confidence in the retrieved trends. The Heat Flux proxy that provides the largest explanatory power in the various fits is a well-known driver of vortex temperature conditions that are the primary causes of polar ozone depletion in periods of high ODS levels. The influence of the GRAD proxy in recent years highlights the importance of the vortex stability for the containment of the ozone hole during the period of maximum depletion"*

Furthermore, we also argued in de Laat et al. [2017] for the use of the ozone mass deficit rather than average ozone or area as the preferred metric to study long term changes in springtime Antarctic stratospheric ozone depletion. The motivation was that the OMD suffers less from what is discussed above (arbitrary choices) then average ozone and area.

As mentioned previously, the motivation of our study was to try and understand the causes of larger ozone variability in the last decade, especially in 2010 and 2012. This is why we chose to base our study on total ozone record. Regarding areas, the use of several thresholds allows us to follow the temporal evolution of areas with low ozone and find possible signs of recovery.

Further, we agree that OMD is a good metric to study the long-term changes. We have thus incorporated this metric using the 220DU threshold in our MLR analyses. Results related to OMD are included in the new Section 5.3 and in Section 6 where we explore the evolution of low ozone values.
The following figure (Figure 11) has been added to the revised version of the paper (Sect. 5.3)

[Figure]

**Figure 11: OMD (in Mt) computed from total columns of MSR-2 dataset lower than 220 DU and south of 60°S for September (left panel) and 15Sept-15Oct (right panel). Regressed values by MLR analysis using GRAD, HF and PWT are also shown as well as the fitted PWT proxy.**

This paper does not address these issues, nor are results put in the context of this work. The paper does show and confirm that most proxies in these multivariate regressions are not really useful. Confirms that statistical significances of post-peak trends become better because of a longer record (but which has to, given the mathematical nature of linear regressions). Confirms that there are differences in trends between September and 15 Sep – 15 Oct. And confirms that there are uncertainties associated with several parameters that need to be defined in advance (vortex position, vortex stability).

But there is no real discussion about why these are the appropriate choices. The GRADS proxy helps in improving the explained variability. But is that the justification? Smaller residuals? If so, I'm sure even better proxies can be constructed.

We have taken into account the legacy of previous works to choose the classical proxies that were used to explain the ozone variability; particularly the work of big-data performed by de Laat et al. (2015). Besides, it is well known that the heat flux (HF) is an important proxy to explain ozone variability. It impacts the evolution of temperature inside the vortex and the build-up of Polar Stratospheric Clouds. But it does not provide an estimation of the vortex permeability and diffusion processes during the period for which the analysis is done. The choice of the GRAD and HF proxies is thus based on physical considerations and not on statistical ones. They allow us to better follow the evolution of the polar vortex on the 1980 – 2017 period. The article was modified in order to highlight those different points in Sect. 5.2.3 as shown in the tracked change version of the manuscript.

Furthermore, the GRADS proxy is detrended. Why? If the GRADS proxy truly represents a physical process, why isn't GRADS allowed to also change on longer timescales (note that this is a point of contention in recent literature: is recovery fully attributable to ODSs or are there other long term changes in atmospheric dynamics that also play a role?).

Both Heat Flux and GRAD proxies were detrended in order to avoid interference to the trend proxy that would be difficult to quantify. Such a treatment is commonly applied to proxy data in MLR analysis. Besides, as shown in Figure 6, not detrending the GRAD proxy would mainly influence the 1980 – 2000 period while the main emphasis of our study is on the recovery period from 2001.
In the paper it is mentioned that our estimation of trend is not necessary due only to ODS.

The same is true for use of the parabolic trend. It is not the standard approach in regression studies (all studies cited use PWLT), but the effect using two linear trends or one parabolic and one linear trends is not discussed (as far as I could see).

We agree with you that a specific discussion was not sufficiently developed on the effect of using a parabolic function in addition to two linear functions instead of two linear functions only for the evaluation of long-term trends. The goal of this additional function is to explain the behaviour of ozone chemical destruction and the effect of saturation of the ozone loss. This was mentioned in the marked-up version of the manuscript on Page 7, line 26 to line 28:

*"In this work our Modified PWLT model (PWT) uses an additional function in order to take into account the slower growth of ODS near the turnaround year and the ozone loss saturation effect within the Antarctic polar vortex in October (Yang et al., 2008)."*

A new Sect. 5.2.4 (PWT vs PWLT) was added to better explain the differences between trends retrieved with our PWT model and the more classical PWLT method.
A figure was added in the supplementary material in order to show that the PWT provides a better representation of long-term ozone evolution within the vortex, especially for the 15Spet.-15Oct. period. Figures S1 and S3 display total ozone inside the vortex and OMD for September and 15Sept-15Oct using the MSR-2 data. The corresponding PWLT and modified PWT regressed values are also shown.

[Figure]

Figure S1 and S3. Top panels: average ozone inside the vortex using the 400 K-600 K range classification for September (left) and 15Sept-15Oct (right). Bottom panels: OMD for both periods. Fitted PWT (black line) and PWLT (green line) are represented in each panel.

It could also be argued based on figure 1 that none of the vortex edge definitions really captures only the vortex core. All still capture some high ozone columns around the vortex edge, which likely introduces variability in the ozone record not related to inner-vortex ozone depletion. Consider that the standard 220 DU value used for OMD and even area fall well inside the 600 K vortex edge.

Our objective was not to capture with our classification the inner vortex only since ozone destruction can occur in the vortex edge in September. The idea was to better constrain the vortex without using any arbitrary ozone-based threshold. As seen in the Fig. 1 of the manuscript, the combination of the different iso-pv lines enables a better selection of the low ozone area. On the particular day shown in the figure, the area selected for the computation of the ozone average is limited by the 400 K vortex line near South America and by the 600 K line on the opposite side. White dot marks were added in the figure in order to better highlight the region selected by the 400 K-600 K range classification method.

This is an exhaustive list of issues, which is exactly the point we want to make here: the issues raised in recent literature about arbitrariness of choices that are made, and the corresponding risk of tunnel vision.

Note that this is also why in de Laat et al. [2017] it is proposed to step away from the whole regression business.

This paper does show that ozone variability is mostly governed by depletion (ODSs) and heat fluxes or vortex (in)stability. How to properly account for the heat fluxes or vortex (in)stability is, however, not really clear, and this paper introduces yet another approach. In de Laat et al. [2017] it is instead proposed to simply

remove the years that are characterized by a more unstable vortex from the record. Such years can be easily identified, but how they affect ozone depletion is much more complex, and appears to depend for example on the exact timing of vortex disturbances [de Laat and van Weele, 2011; 10.1038/srep00038]. This paper provides some additional ammunition for the proposal to step away from the regression methods.

In our work, we used another approach and tried to reproduce the ozone variability for all years of the studied period. As mentioned previously the objective was to try and explain ozone variability in the last decade

The presence of this exhaustive list of issues and questions would be less of a problem if the paper introduced new concepts or new ideas, but the paper mostly builds on previous work and confirms what other papers have also concluded.

The new concepts and ideas that are introduced in the paper do not help in clarifying in what has recently emerged: the sometimes large structural uncertainties in this particular field of research, and arbitrariness with which analyses are performed. If anything, they only confirm the existence of large structural uncertainties and the arbitrariness.

We hope that the many arguments we have developed previously help explain the contributions of our paper. Despite structural uncertainties in the MLR technique, it is widely used in ozone and climate studies and the level of agreement with observations obtained with our model gives us some confidence in our results. Further, your review allowed us to substantially improve the article and better explain our approach

So, how do I think this paper could be improved?
[1] One possibility would be to include additional analyses cover more of the parameter space. The paper already also uses average ozone and area, so a mass deficit could be included as well (see Fig. 5 of de Laat et al. [2017]).

The OMD was included in the MLR analysis using for a threshold of 220 DU, south of 60°S. For simplicity, the OMD is computed for the periods of our study, e.g. September and 15Sept-15Oct. The OMD was also evaluated for different thresholds in order to compare with our evaluation of ozone hole areas with low ozone values.

The use of different area definitions based on different ozone thresholds could also be expanded – like looking at changes in the probability distributions of total ozone (a bit like Yang et al. [2008; 10.1029/2007JD009675], but much more extensive). However, that would require a considerably amount of additional work.

We have added two additional thresholds (175 DU and 200 DU) to refine the study. A choice of alternative thresholds as a function of probability distributions of total ozone could be done in a future work.

I could live without such an analysis if:

[2] regardless, results should be discussed within the context of recent publications and criticism of existing methods of Antarctic stratospheric ozone recovery detection. This is currently lacking, as also reflected in the conclusions section, which is more of a summary than a conclusion.

This issue was revised in the paper. For example in the introduction section (page 2, line 18 to line 22)

*"The limitation of MLR analysis is that only formal statistical error of trend is estimated and structural uncertainties linked to the single and arbitrary combination of proxies is not taken into account. De Laat et al. (2017) inferred trend values from daily Ozone Mass Deficit (OMD) computed from a multi-sensor reanalysis dataset without using any model but filtering the anomalous years with low polar stratospheric cloud (PSC) volume. The authors found positive and highly significant trend of OMD since 2000."*

The conclusions were also modified accordingly (page 14, line 26 to line 30).

*"The structural uncertainties of the MLR analysis linked to the selection of proxies were not fully analysed in this work, as in De Laat et al. (2015). The main sensitivity tests concerned the baroclinicity of the vortex and the impact of its stability during the studied periods. Trend differences in the various scenarios analysed provide some quantification of related uncertainties and are lower than the statistical trend uncertainties. Further, the large determination coefficients obtained for both periods analysed give confidence in the retrieved trends."*

The challenge here will be to discuss it in such a way that that discussion does not undermine the findings of the paper.

So, what should be discussed are what I consider the most important findings of the paper:

- Most proxies in the MVR do not contribute much (if anything) to reduce trend uncertainties (small explanatory power)
- September yields a higher statistical trend significance than 15 Sep-15 Oct.
Those two points are more emphasized in the text and especially in the conclusions. For example in page 13, line 27 to line 37 about the different proxies:
 *"While the HF combined with GRAD proxies reproduce quite well the interannual variability of ozone, other proxies such as Aerosols, QBO, SF and AAO present smaller explanatory power and contribute less to reduce trend uncertainties."*

- Range of trend values and trend significance levels are indicative (or not) for structural uncertainties and systematic errors (this needs to be further supported)
See our answer above

In addition, I think the following should be included in a revision:

- report 2000-2010 and 2000-2012 trends & statistics for comparison with the 2000-2015 trends (and significances). This is helpful for comparison with results from older previous papers using MVR methods but somewhat different proxies.
A sensitivity test was made on the length of the datasets after 2001 varying the end year from 2010 to 2017 using PWT and PWLT proxies. The PWLT proxy has shown higher positive trends for the recovery period for both months compared to the trends based on PWT MLR analysis. PWLT presents significant trends but for PWT the significance of trends for the 15Sept-15Oct depends on the end year. As expected, error bars are smaller depending on the length of the dataset.

The 2001-2010 and 2001-2012 results were compared to previous works using PWLT proxies since PWT was never used before. A significant trend of 4.15 DU $yr^{-1}$ is found in September, higher than 3.3 DU $yr^{-1}$ reported in de Laat et al (2015) for Sept-Nov period and turnaround year in 2000. The $R^2$ coefficient is higher than 0.87 for all cases. Since trend proxies and ozone period are different, a comparison with previous results for the 2001-2010 and 2001-2012 was no introduced in the paper. A comparison with Solomon et al results for September was however included.

- Use of "area" for 150 DU of 125 DU is an interesting more or less novel approach. Results show that such small TCOs did not occur until the late 1980sand early 1990s, indicative that these parameters are more sensitive for more severe ozone depletion. This also means that these parameters should return back to zero values earlier than the TOC columns return to 1980 levels. This method/analysis could be expanded more, by using the 150 or 125 DU also as vortex edge proxies (average ozone within area), and for Ozone Mass Deficit calculations (which traditionally is based on the 220 DU level but that is somewhat arbitrary). Possibly also report 175 and 200 DU results.
Additional thresholds were considered as suggested. Our study is now based on 125, 150, 175, 200 and 220DU (Fig. 12). Such thresholds were also included in the analysis of OMD evolution.

(in all honesty, I think the analysis of long term changes in probability distributions could be a topic of a completely separate paper)
It is a very interesting suggestion and we will consider it for a future work.

**3. Reply to minor comments**

Page 1, line 25-26, and line 29 (and correspondingly tables 2 & 3), in particular the range of trend values that are reported.
How should this range be interpreted? Could this be considered representative of the structural uncertainty?
Since a small number of cases were considered in our study to evaluate structural uncertainties, this range can be interpreted in a qualitative way. A discussion in the Conclusion section about the comparison of extreme trend values found for the different cases treated in the paper at each period and the formal statistical error bars was performed. In September, the range of extreme cases is within the 2 sigma statistical error bars, while it is higher for the 15Sept-15Oct period.

Page 2, line 14-15, the explanation of why October ozone behaves differently from September ozone.
October ozone is partly governed by different processes than September ozone. First of all, catalytic photochemical ozone destruction ceases in October. Rather, there is regeneration of ozone due to photolysis of O2 and oxidation of CH4 and carbon monoxide [Grooss et al., 2011; 10.5194/acp-11-12217-2011]. Furthermore, there is continued downward transport of ozone rich outer-vortex air into the vortex from the upper stratosphere down to the lower stratosphere [de Laat and van Weele, 2011; doi:10.1038/srep00038]. And there is vortex dynamics, as the authors correctly note. Together, these processes to a large extent determine October Antarctic inner vortex ozone.
We are aware of the processes governing the ozone levels in October (see previous publications of our team Godin et al., J. Geophys. Res., 106(D1), 1311-1330, 2001 and Pazmiño et al, Atmos. Chem. Phys., 8, 5339–5352, doi:10.5194/acp-8-5339-2008, 2008 which analysed some of them). In our study, we focused on the baroclinicity of the vortex linked to vortex dynamics. We agree that it is not the only process affecting total ozone levels. The sentence was thus changed as follows (page 2, line 26 to line 28):
*"The baroclinicity of the polar vortex in October and its displacement from the geographic pole can also contributes to the variability of the total ozone series averaged during the month of October."*

Page 4, line 15. It is stated that a 5-day smoothing is applied to the EL of the maximum PV gradient. However, as far I know Nash et al. [1996] does not call for a 5-day smoothing. If that is right, then what is the justification of the 5-day smoothing?
A justification of this smoothing has been added (page 4, line 34 to line 35):
*"This limit is subsequently smoothed temporally with a moving average of 5 days to reduce the noise in the vortex edge data series."*

Page 5, line 6-7. Correlations. Sometimes the paper uses R, sometimes R2. Be consistent, preferably using R2 and only refer to R if the correlation is negative (still providing R2).
See also: page 5 - line 16, Page 10, line 17, and make sure to check throughout the paper.
We decided not to follow your suggestion. We prefer to use a correlation coefficient R when comparing data records and a determination coefficient $R^2$ when comparing measurements with regression model retrievals.

Page 5, figure 4. The differences between SAT and MSR2 are fairly straight forward to explain. Up until 1993, both rely solely on TOMS. From 1993-1995, MSR2 relies on SBUV, and thanks to the data assimilation gaps are filled. From 1996 onwards, MSR-2 also uses GOME (1996 to 2005), SCIAMACHY (2002-2012), OMI (2004-), and GOME2 (2007-). Furthermore, MSR-2 uses ground-based total column data to account for inter-instrument differences. As a result, the estimated average MSR-2 total ozone column bias has been estimated at 1% [van der A et al., 2015; amt-8-3021-2015].
Please note that figure where changed since SBUV data was replaced by MSR-2 considering the spatial coverage of TOMS/OMI.

Add to line 21 the following "whole vortex. The data assimilation of MSR2 to some extent does fill gaps when ozone measurements are limited."
Add after line 25. These differences are caused by MSR-2 starting to use multiple satellite total ozone column records after 1996, the procedures in MSR-2 to account for inter-instrument differences, and the data assimilation methodology that allows for filling gaps [van der A et al., 2015].
This part was changed (page 6, line 1 to line 19)

Page 6, line 25. It is stated that both PWLT and a combined parabolic trend – linear trend is generally used. The latter is not true, all papers cited only rely on a PWLT. The parabolic trend is a new concept introduced in this paper. As such, it should be explained later in the paper what the differences are that associated with both PWTs (the PWLT appear no to be used in the paper at all)

You are right, previous papers use only PWLT. A new section, the Section 5.2.4 (PWT vs PWLT) was added to the revised version of the paper.

Grammar, typos.

Page 1, line 27. Replace "lower than" with "smaller than"  done

Page 2, line 4: change to "interannual variability of ozone as a function of the 11 year"  done

Page 2, line 8. I assume what is meant is "for the period over which the ozone record is calculated and for …" Yes. Done

Page 2, line 12. "ozone content is deepest", I think what is meant here is "where ozone depletion is largest" or "where the ozone deficit is largest". The sentence was changed accordingly

Page 2, line 19. "update of the ozone" done

Page 2, line 22. "full development of Polar ozone depletion". I think what is meant here is "the period of fastest catalytic photochemical ozone destruction" The sentence was modified accordingly

Page 3, line 35. Include reference to de Laat et al. [2017; 10.1002/2016JD025723] as a paper that also uses MSR2. done

Page 4, line 15. Change to "This limit is subsequently smoothed temporally with" done

Page 4, line 17. Start with "The Nash criterion" done

Page 4, line 29. Change to "On this particular day, the region …" done

Page 4, line 32. Change to "consist of" done

Page 4, line 35. Change to "using the new classification." done

Page 4, line 36. Change to "The standard classification estimates a 40 DU and 20 DU larger ozone mean …" done

Page 5, line 3. Change to "for the SAT data series … … based on the single …" done

Page 5, line 4. Change to "Error bars represent the two sigma …" done

Page 5, line 7. Change to "at the 2σ level" done

Page 5, line 12. Change to "is preferred since it takes …" done

Page 5, line 35. Change to "The ODS contribution to long-term trends in ozone is represented by piece-wise linear trend …" done

Page 6, line 15. Start new paragraph after "period" done

Page 6, line 21. Change to "with a p-value" done

Page 7, line 14-17. Rephrase line "Despite … Weber et al. 2017)". I assume you want to note that although September shows large variability in total ozone, it is still a commonly used month for recovery detection. Yes you are right. The sentence was removed and replaced in Page 8, line 22 to line 23 by

*"Although pronounced decrease in total ozone is observed in September, recent works have used ozone records obtained during this month to detect the ozone recovery (Solomon et al., 2016; Chipperfield et al., 2017; Weber et al., 2017)."*

Page 7, line 18. Remove "are highlighted", change "conclude that" to "identify " done

Page 7, line 18. Change "on October" to "for October" done

Page 7, line 20-21. Delete "In our study … previous section." done

Page 7, line 25. Change to "the year 2000 was characterized by …" done

Page 7, line 26. Change to "September, and yields a relatively high …" done

Page 8, line 7. Add reference to Chipperfield et al. [2017; doi:10.1038/nature23681], who amongst others discuss the differences in pre-post peak ozone recovery rates. done

Page 9, line 30. Change to "at 550K where the trend after …" this sentence was removed

Page 9, line 37-38. Change to "Trends estimate for the second period show slightly" This sentence was removed

Page 9, lines 40-41. Please rephrase, I don't fully understand what is meant here. The sentence was modified (page 10, line 18 to line 23)

*"Despite the good agreement between regressed values and measurements especially for the period 15Sept-15Oct and for the range classification method (400 K-600 K), it is not possible to attribute ozone significant increase to ODS decrease. In addition, the ratio between trends before and after 2001 is larger than 3-which could be due to the effect of desaturation of the ozone loss."*

Page 10, line 1. Change to "higher than 3, the threshold value …" sentence modified, see previous point (page 9 of original version)

Page 10, line 21. Change to " The ozone hole is also frequently defined as …" We prefer to write "generally" instead of "also frequently"(page 9 of original version)

Captions of figure 11 + 12: OMIT $\Rightarrow$ OMI
Previous Fig. 11 and 12 were modified. The Fig 11 is the new Fig. 12 and Fig 12 is the new Fig. 13 in the revised version and only MSR-2 data was used.

**Reply to Anonymous Referee #2 review of manuscript acp-2017-1157**

**Symptoms of total ozone recovery inside the Antarctic vortex during Austral spring**

Andrea Pazmino on behalf of all co-authors

We thank Anonymous Referee #2 for the time devoted to evaluate our work. Your valuable comments have helped us to improve our manuscript. Since MSR-2 total ozone data have become available until the end of October 2017, we decided to extend our study to the year 2017 using SAT and MSR-2 data. Due to this extension, all figures of the manuscript have been revised, except Figure 1 where white cross marks where added to highlight the region considered inside the vortex by the 400 K-600 K classification range. In addition we have noticed that the figure 12 of the original manuscript about the time shift of low values was not very clear. A new figure, Figure 13, has been produced in order to better illustrate the time shift in appearance of low total ozone values within the vortex. Similar conclusions as in the original version of the manuscript were provided. Furthermore the word "Multiple" was added to the title to highlights that different signs of recovery were obtained in this work, , e.g. (1) Significant positive trends of total ozone since 2001 in September and for the first time in the period of maximum ozone depletion (15Sept-15Oct) using MLR analysis on average ozone inside the vortex and Ozone Mass Deficit, (2) Decrease of occurrences of very low ozone values within the vortex and (3) increased delay of occurrence of low total ozone levels in the September 1st – October 15th period.

Please find our answers to your comments (in red):

I concur with much of what the other reviewer articulated, in particular these points from DeLaat's review:
1. "The presence of this exhaustive list of issues and questions would be less of a problem if the paper introduced new concepts or new ideas, but the paper mostly builds on previous work and confirms what other papers have also concluded."

2. "This paper does not address these issues, nor are results put in the context of this work."

3. "The few time series that are looked at are then seen as the truth, every wiggle becomes meaningful, and too much attention is given to the formal statistical significances, whereas structural uncertainties are important as well. For example, we have shown that rather arbitrary choices with regard to the proxies used in the regression have a strong impact on the formal statistical trend errors. We therefore argued that structural uncertainties are much larger than the formal statistical trend errors, which is important for confident statements about whether recovery has started or not."

I especially agree with DeLaat's concerns about the 'structural uncertainties' in this regression analysis, so please address all issues described in his review. In addition, there are other issues below related to ozone data sets that need to be addressed in a revised manuscript. If revisions are made that address both DeLaat's and my review, this paper could be published in ACP.

Please see our answer to de Laat's review to the different points specified above.

**Specific topics of Concern**

The composite satellite total ozone time series, referred to as SAT. The merging of satellite data sets into a single record is something to be done very carefully. Instrument measurements have bias and drift, and combining data sets in order to extract small trends (i.e., ozone recovery) requires a great deal of care and a good deal of knowledge about each instrument's characteristics and sampling pattern (i.e., coverage). I see no evidence here that any such considerations were used when combining the data sets. In fact in Figure 4, the difference between the assimilated ozone time series (MSR) and the SAT shows big jumps! There is a large trend from 1990-2005. Does this represent an unphysical trend (i.e., changes in the observing system) in the assimilation, or is this coming from how the individual data sets in the SAT were merged? Have you tried your trend analyses on the 5 merged ozone data sets referenced in Weber et al. [2017]? Without any discussion or justification of how the data sets were merged in this study, I don't see how the trend results presented here (and especially their uncertainties!) can be taken seriously.

Since MSR-2 data are based on the assimilated satellite ozone time series already corrected from offset, trends and variations of solar zenith angle and temperature in the stratosphere, we would like to consider in addition in our work a satellite datasets commonly used in ozone studies (including recovery) with similar kind of instrument (TOMS and OMI) and similar retrieval; without applying any correction. Since SBUV data are sparse, we have decided in the revised version to fill the 1993 – 1995 gap years with MSR-2 data, but taking into account the same spatial coverage as that of TOMS and OMI instruments (see new Fig. 4). Finally, due to the important differences observed in September, particularly the unexplained trend in the 1990-2005 period that you mention, we decided not to include SAT datasets for trend retrieval in that month. Discussions about this difference as well as the issue of spatial coverage of the SAT data is discussed in the Sect. 4 of the marked-up manuscript. The following paragraph was added in page 6, line 1 to line 19

*"MSR-2 total ozone data series inside the vortex are compared to SAT series as shown in Fig. 4, which displays the relative difference between MSR-2 and SAT for the 400K-600K range classification. Differences of about ±0.5% are observed in the 1980s. Small differences are expected during this period since only TOMS data are used in both data sets until 1993. In the 1993-1995 period discrepancies between both curves are only due to the differences in the selection of MSR-2 data for the SAT record in order to have similar spatial coverage as the data from the other instruments incorporated in the SAT time series. These differences varying between -1 and 0.5 % represent an estimation of the impact of reduced spatial coverage in SAT dataset on the averaged total ozone level in September. The 15Sept-15Oct period presents negligible differences. The addition of GOME (1996-2005) in MSR-2 assimilation could explain the discrepancies with the SAT dataset that considers only TOMS-EP. From 2001, differences are larger and generally positive, reaching ~5% in September and ~3% in 15Sept-15Oct. period. These increased differences are especially visible during the period where data from instruments on board the ENVISAT platform (e.g. SCIAMACHY) are assimilated in the MSR-2 record. Overall, values in September present a mean bias of 1.3 % (dash blue line in Fig. 4), and in 15Sept-15Oct a smaller bias value of 0.5 % (dash red line in Fig. 4). Temporal evolution of the differences, e.g. negative trend in the 1980s and positive trend in the 2000s, can have an impact on the long-term ozone trends retrieved from both records. In general, differences between SAT and MSR-2 records are caused by MSR-2 starting to use multiple satellite total ozone columns records after 1996, the procedures in MSR-2 to account for inter-instrument differences, and the data assimilation methodology that allows for filling gaps (van der A et al., 2015)."*

Regarding the 5 merged ozone data sets referenced in Weber et al. [2017], they correspond to zonal averages and cannot be used for total ozone classification as a function of equivalent latitude as it is done in our study.

The 'range method' is not clearly explained. I understand that you are using it to see the sensitivity of the calculated trends to the definition used for the area of depletion, and I get that you calculate different areas depending on which isentropic level is used, but exactly how are you deciding which levels to use? Are you averaging over all the 400-600K level results? Only some of them? Do you choose the same range for each year? The details of this methodology were not made clear. It's interesting that in the end you conclude that the 475K results are as good as the other definitions. Is this because this is an altitude where there is some of the most severe depletion? An explanation for this result should be offered.

The range method was better explained in Sect. 3.2 (Methodology for classification), in Page 5, line 4 to line 8.

*"The total ozone column may thus not represent the ozone behaviour inside the vortex. In order to consider possible vortex baroclinicity, another approach is used, where vortex classification at different isentropic levels is considered at the same time. For this second approach, the range of selected isentropic levels is chosen in the altitude region of maximum ozone depletion: from 400 K to 600 K with a step of 25 K. The same 9 isentropic levels considered for 400 K-600 K range classification are applied each year."*

The range classification considers selected isentropic levels between 400 K and 600 K with a step of 25K. Then the same 9 isentropic levels are used each year for the classification. While this new classification provide a better constraint of low ozone values within the vortex, differences in trend results are not significant at 2 sigma levels. We suggest in the revised version that the reason could be the good correlation between the different data sets (R>0.98) using the different methods. Sentences in the Conclusions (page 13, line 15 to line 24) were changed as follow:

*"For the classification of total ozone measurements inside the vortex, the classical Nash et al. (1996) method is used. In order to evaluate the impact of vortex baroclinicity on trend analysis, classifications using a single isentropic levels (475 K, 550 K) and a range of levels (400 K – 600 K) are tested. Systematic differences are found between the various total ozone time series. However the inter-annual variability is similar with correlation coefficients ranging from 0.98 to 0.99 in both studied periods. While larger trend values are generally found with the 475 K classification, the differences with trends related to the 400 K – 600 K range classification are not significant at $2\sigma$ level."*

The satellite instruments used (all UV sensors) do not see to the south pole in early September. The analysis calculated results for the polar region for the entire month of September, but measurements cannot be made at the highest latitudes in early September. Thus the 'September average' will be more strongly weighted by lower latitudes and later September dates. Please describe how the satellites' sampling of the polar area varies over September and what this does to the 'September averaged' quantity. This may impact the meaning of the trend results as they will include more of the late September, higher dynamical variability measurements.

We agree with your arguments. We have excluded SAT datasets for September also for this reason. In addition we have included a sentence on UV sensors sampling in September where measurements are not available for regions poleward of 77°S in the beginning of September, 82°S mid-September and 89°S at the end of the month in the Sect. 2.1 of the marked-up manuscript (page 3, line 31 to line 33)

*"Since TOMS and OMI UV sensors do not receive enough UV light in early September, originating from regions not illuminated by the Sun (from 77°S to 82.5°S up to mid-September), these regions were not considered to compute the total ozone mean value in MSR-2 data."*

The impact of spatial coverage differences between SAT and MSR-2 was discuseed in Sect. 4 (page 6, line 6 to line 10).

[revised manuscript text omitted]

---

## Referee Report (RR1)

Review of manuscript acp-2017-1157 (revision)

Multiple symptoms of total ozone recovery inside the Antarctic vortex during Austral spring.

Pazmino et al., 2018.

**Review**

The changes and modifications that have been made to this paper have very much improved the paper, and I thank the authors for putting in considerable effort for making improvements. I am quite pleased with the paper as it is right now: it is more consistent and more complete, and provides a solid contribution to the ongoing debate on ozone recovery in the Antarctic ozone hole.

There are two minor issues left, as well as some typos and grammar that need to be addressed, see below.

**Minor issues**

[1] Conclusions: I would invite the authors to spend a few sentences on what their take is on the attribution question, i.e. how much of the increase in Antarctic ozone can be attributed to changes in ODSs and how certain are they. The authors are free to provide their views, but the reason to ask is that the debate about Antarctic ozone recovery has changed from the question whether recovery is observed (which is now well established) to how much of the observed increase in Antarctic ozone can be attributed to the decrease in ODSs. All of it, all of it but uncertainties cannot exclude the possibility that part of it is not related to ODSs, only part of it, etc. That way the paper contributes to this discussion, which no doubt will take some time to settle.

[2] Page 12, last paragraph. It is argued that the observed 2001-2017 reductions in OMD (53% and 35%) are too large compared to the decrease in ODS during the same time period. However, this is an incorrect interpretation.

The change in OMD is relative to the arbitrary 220 DU level. The 220 DU OMD only starts to deviate from zero around 1980 (ozone columns smaller than 220 DU did not systematically occur before that time). So, the change in fraction change in OMD should be compared to the fraction change in ODSs between 2001 and 2017 RELATIVE to the 1980 ODS levels.

According to figure S1-1 of the 2010 Ozone Assessment, 1980 ODS levels were 50% of what they were in 2000. The corresponding change in OMD 2000-1980 has been approximately 20-25 Mt. Between 2001 and 2017, ODSs have decreased by 15-20%, which is 30-40% of the change in ODS since 1980, and thus as 30-40% change in OMD since 1980. Which is consistent with the numbers reported (53% and 35%).

ODSs were note zero in 1980, and as others have shown, ozone depletion really started already after 1960 [Langematz et al., 2016; https://doi.org/10.5194/acp-16-15619-2016].

Please modify this paragraph accordingly.

**Typos, grammar**

General: make sure to replace "higher" with "larger" throughout the paper where applicable. I know it is acceptable to talk about for example "higher values", but in the context of the atmosphere and especially in a paper where heights play a role, I strongly prefer the use of "larger" over "higher", as the use of "higher" can be confusing.

Page 2, line 1. Replace "deeply" with "strongly"

Page 2, line 23. "evaluated ozone trends using" (plural trends)

Page 2, line 23. Specific which ozone trends Solomon et al. [2016] have evaluated (both total ozone and ozone profiles/height resolved ozone)

Page 2, line 24. Replace "The authors have shown a significant" with "They found a significant"

Page 2, line 30. Replace "remains thus an open question" with "remains open to debate"

Page 4, line 39 (+ line 1 page 5). "The white dot marks in figure 1" (remove "the")

Page 5, line 1. A dot is missing after "classification".

Page 5, line 4. Delete "isentropic levels" (is already mentioned before in the same sentence)

Page 7, line 33. Delete "of"

Page 7, line 34. Change to "… September, with very low values observed mostly during the last week."

Page 7, line 35; Page 8, line 3. Suggest to replace "works" with "publications"

Page 8, line 2. "In this paper, …"

Page 8, line 29. Replace "only" with "alone"

Page 9, line 14.  Replace "more" with ""better"

Page 9, line 15. Delete "thus for the first time". This is not true, in our 2017 paper we already discuss trend significances for a whole range of time periods, and we did find a significant trend also for the period 15 Sep – 15 Oct albeit less significant than periods including early September.

---

## Author Response (AR2)

**Reply to A. T. J. de Laat review of revised manuscript acp-2017-1157**

**Multiple symptoms of total ozone recovery inside the Antarctic vortex during Austral spring**

Andrea Pazmino on behalf of all co-authors

We thank again A. T. J. de Laat for the thorough review of our revised version of the manuscript. Please find our answers (in red)

**Comments to review**

The changes and modifications that have been made to this paper have very much improved the paper, and I thank the authors for putting in considerable effort for making improvements. I am quite pleased with the paper as it is right now: it is more consistent and more complete, and provides a solid contribution to the ongoing debate on ozone recovery in the Antarctic ozone hole.
There are two minor issues left, as well as some typos and grammar that need to be addressed, see below.

**Minor issues**
[1] Conclusions: I would invite the authors to spend a few sentences on what their take is on the attribution question, i.e. how much of the increase in Antarctic ozone can be attributed to changes in ODSs and how certain are they. The authors are free to provide their views, but the reason to ask is that the debate about Antarctic ozone recovery has changed from the question whether recovery is observed (which is now well established) to how much of the observed increase in Antarctic ozone can be attributed to the decrease in ODSs. All of it, all of it but uncertainties cannot exclude the possibility that part of it is not related to ODSs, only part of it, etc. That way the paper contributes to this discussion, which no doubt will take some time to settle.

Many papers especially published since 2016 have indeed shown an ozone recovery in the Southern Polar Regions. On the other hand, our revised paper assesses the last stage of ozone recovery using different methods: MLR analysis of total ozone and ozone mass deficit in two different periods. It is the first one to demonstrate a significant increase in total ozone in the period of maximum ozone depletion within the polar vortex and to show a decrease in low ozone values within the vortex in the period September $1^{st}$-October 15. It addresses also the issue of later appearance of ozone depletion already mentioned by Solomon et al., 2016. As you mention, the main issue now is to estimate the contribution of ODS decrease in the observed ozone increase since the beginning of 2000s. It is clear that MLR analysis alone cannot address this issue and studies combining model and observations are needed for that. We have added the following sentences in the conclusion (page 12, line 34 to 40)

*"However, as for other trend studies based on MLR fit to observations, it is not possible from this analysis alone to fully attribute the retrieved trends to ODS evolution. For such a study, a combination of model and observations is needed. Potential feedbacks between chemistry, radiation and dynamics will play a role in ozone recovery. A recent study indicates an increase in temperatures within the vortex core from MERRA reanalyses during the period 2000 – 2014 in austral spring and summer (Solomon et al., 2017). Such a temperature increase that could be linked to ozone increase could play a role in the decrease of occurrence of low ozone values within the vortex and subsequent ozone increase"*

Reference to Solomon et al. (2017) was also added.
*"Solomon, S., Ivy, D., Gupta, M., Bandoro, J., Santer, B., Fu, Q., Lin, P., Garcia, R. R., Kinnison, D. and Mills, M.: Mirrored changes in Antarctic ozone and stratospheric temperature in the late 20th versus early 21st centuries, J. Geophys. Res. Atmos., 122, 8940–8950, doi:10.1002/2017JD026719, 2017."*

[2] Page 12, last paragraph. It is argued that the observed 2001-2017 reductions in OMD (53% and 35%) are too large compared to the decrease in ODS during the same time period. However, this is an incorrect interpretation.

The change in OMD is relative to the arbitrary 220 DU level. The 220 DU OMD only starts to deviate from zero around 1980 (ozone columns smaller than 220 DU did not systematically occur before that time). So, the change in fraction change in OMD should be compared to the fraction change in ODSs between 2001 and 2017 RELATIVE to the 1980 ODS levels.

According to figure S1-1 of the 2010 Ozone Assessment, 1980 ODS levels were 50% of what they were in 2000. The corresponding change in OMD 2000-1980 has been approximately 20-25 Mt. Between 2001 and 2017, ODSs have decreased by 15-20%, which is 30-40% of the change in ODS since 1980, and thus as 30-40% change in OMD since 1980. Which is consistent with the numbers reported (53% and 35%).

ODSs were note zero in 1980, and as others have shown, ozone depletion really started already after 1960 [Langematz et al., 2016; https://doi.org/10.5194/acp-16-15619-2016].

Please modify this paragraph accordingly.

*You are right on this point. The paragraph was modified accordingly (page 13, line 2 to 4):*

*"Similar reductions of 53 % and 35 % of OMD are computed for September and 15Sept-15Oct respectively. This is consistent with the 30 % - 40 % change in ODS relative to their level in 1980 when total ozone values below the 220 DU threshold started to appear systematically (WMO, 2011)."*

**Typos, grammar**

General: make sure to replace "higher" with "larger" throughout the paper where applicable. I know it is acceptable to talk about for example "higher values", but in the context of the atmosphere and especially in a paper where heights play a role, I strongly prefer the use of "larger" over "higher", as the use of "higher" can be confusing.

Thank you for the remark. The text was revised accordingly

The following remarks were also considered:
Page 2, line 1. Replace "deeply" with "strongly"
Page 2, line 23. "evaluated ozone trends using" (plural trends)
Page 2, line 23. Specific which ozone trends Solomon et al. [2016] have evaluated (both total ozone and ozone profiles/height resolved ozone)
Page 2, line 24. Replace "The authors have shown a significant" with "They found a significant"
Page 2, line 30. Replace "remains thus an open question" with "remains open to debate"
Page 4, line 39 (+ line 1 page 5). "The white dot marks in figure 1" (remove "the")
Page 5, line 1. A dot is missing after "classification".
Page 5, line 4. Delete "isentropic levels" (is already mentioned before in the same sentence)
Page 7, line 33. Delete "of"
Page 7, line 34. Change to "… September, with very low values observed mostly during the last week."
Page 7, line 35; Page 8, line 3. Suggest to replace "works" with "publications"
Page 8, line 2. "In this paper, …"
Page 8, line 29. Replace "only" with "alone"
Page 9, line 14. Replace "more" with ""better"

The remark below was not considered since de Laat et al. (2017) discussed about trends significance of OMD and not total ozone as we mention in Sect. 5.2.2 for the period of maximum ozone depletion (page 9, line 1 to 28)

Page 9, line 15. Delete "thus for the first time". This is not true, in our 2017 paper we already discuss trend significances for a whole range of time periods, and we did find a significant trend also for the period 15 Sep – 15 Oct albeit less significant than periods including early September.

[revised manuscript text omitted]